# OrtSAE: Orthogonal Sparse Autoencoders Uncover Atomic Features

## Abstract

Sparse autoencoders (SAEs) are a technique for sparse decomposition of neural network activations into human-interpretable features. However, current SAEs suffer from feature absorption, where specialized features capture instances of general features creating representation holes, and feature composition, where independent features merge into composite representations. In this work, we introduce Orthogonal SAE (OrtSAE), a novel approach aimed to mitigate these issues by enforcing orthogonality between the learned features. By implementing a new training procedure that penalizes high pairwise cosine similarity between SAE features, OrtSAE promotes the development of disentangled features while scaling linearly with the SAE size, avoiding significant computational overhead. We train OrtSAE across different models and layers and compare it with other methods. We find that OrtSAE discovers $9\%$ more distinct features, reduces feature absorption (by $65\%$) and composition (by $15\%$), improves performance on spurious correlation removal ($+6\%$), and achieves on-par performance for other downstream tasks compared to traditional SAEs.

## 1 Introduction

Large Language Models (LLMs) have achieved remarkable performance in natural language processing, but their internal mechanisms remain poorly understood. Mechanistic interpretability aims to understand how neural networks function by reverse-engineering their computational processes (Olah et al., 2020). Central to this field is understanding *features*, the human-interpretable concepts represented as directions in a model's internal representation (Elhage et al., 2022; Park et al., 2023).

Early interpretability methods focused on analyzing individual neurons (Olah et al., 2020; Bills et al., 2023), but a key challenge has been that neurons are often *polysemantic*, responding to multiple unrelated concepts rather than encoding single interpretable features (Olah et al., 2020). One theory of why polysemanticity occurs is *superposition*, which posits that neural networks represent more features than they have dimensions (Elhage et al., 2022). Although this enables efficient use of model capacity, it significantly complicates interpretability research.

Sparse Autoencoders (SAEs) have emerged as a powerful approach to disentangling superposition (Bricken et al., 2023; Cunningham et al., 2023). By adding a sparsity penalty to the reconstruction loss, SAEs learn to decompose activations into a sparse latent space where each dimension aims to capture a distinct, interpretable feature (Gao et al., 2024; Marks et al., 2024). Traditional SAE variants (Bricken et al., 2023; Gao et al., 2024; Bussmann et al., 2024; Rajamanoharan et al., 2024) focused on improving reconstruction quality while maintaining sparsity. However, the standard objective can lead to two failure modes. As the number of SAE latents grows, *feature absorption* can occur (Fig. 2a), where a broad feature representation absorbs into more specific, token-aligned latents (e.g., a latent "starts with E" will activate on all tokens starting with "E", except for the token "elephant") (Chanin et al., 2024). Another issue is *feature composition* (Fig. 2b), in which independent features (e.g. representing "red" and "square") are merged into a single composite feature ("red square") (Leask et al., 2025). Both problems undermine the interpretability of SAE latents and the applicability of SAE representations for downstream tasks (Karvonen et al., 2025). To address these issues, Bussmann et al. (2025) introduced Matryoshka SAE, a hierarchical approach to organizing features at multiple levels of abstraction. However, this method introduces additional computational

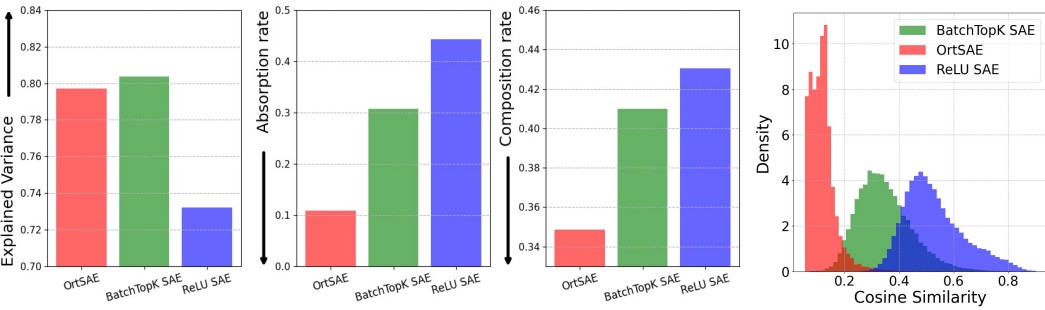

Figure 1: **Performance of OrtSAEs vs. traditional SAEs.** Bar plots display explained variance, absorption, and composition rates for three SAE variants at L0=70 sparsity. OrtSAEs show a marginally lower explained variance than BatchTopK SAEs but decreased absorption and composition, indicating better feature specificity. The density plot illustrates the distribution of pairwise cosine similarity values, computed as the maximum similarity between each decoder feature and its closest counterpart in the model, across all features at L0=70. OrtSAEs demonstrate lower pairwise cosine similarity, confirming greater decoder feature orthogonality compared to BatchTopK and ReLU SAEs.

overhead and suffers from feature hedging (Chanin et al., 2025), a problem where correlated features merge at higher levels, reducing interpretability. This highlights the need for alternative approaches.

Feature absorption and composition lead to redundant representations where multiple latents capture overlapping concepts, which results in high cosine similarities between them. This suggests that enforcing orthogonality between SAE latents could provide a principled approach to mitigate these issues. Therefore, we propose **OrtSAE**, a novel approach to SAE training that promotes the emergence of more atomic features (Sec. 3.3). At each training step, we penalize high cosine similarities between SAE latents by introducing an additional orthogonality penalty. To optimize computation, we implement a chunk-wise strategy that divides SAE latents into smaller blocks, computes the penalty separately, and aggregates the results. This reduces the complexity from quadratic to linear with respect to the number of latents and introduces a negligible computational overhead. Importantly, this penalty scales efficiently without altering the core SAE architecture.

We train OrtSAE on the Gemma-2-2B (Team et al., 2024) and Llama-3-8B (Dubey et al., 2024) and compare it against traditional SAEs and Matryoshka SAE (Bussmann et al., 2025). Experimental results demonstrate that our objective reduces feature absorption and composition across a wide range of sparsity levels (Sec. 4.3). For example, at a L0 of 70 (Fig. 1), OrtSAE discovers $9\%$ more distinct features, reduces feature absorption by $65\%$, and feature composition by $15\%$ compared to traditional SAEs. On SAEBench (Karvonen et al., 2025), our method improves performance on spurious correlation removal by $6\%$ while maintaining on-par performance for other downstream tasks (Sec. 4.4). Through qualitative experiments, we show that OrtSAE features efficiently decompose composite features learned by other SAEs into more atomic components (Sec. 4.3).

Our paper makes the following contributions:

- We propose OrtSAE, a novel approach to SAE training that directly addresses the issues of feature absorption and composition, without requiring complex architectural changes or significant computational overhead (Sec. 3.3).

- Comparison of OrtSAE with traditional SAEs shows that our method produces more distinct features, reduces absorption and composition rates (Sec. 4.3).

- Experimental results on SAEBench demonstrate that our method performs on-par with other SAE architectures, and outperforms them on spurious correlation removal (Sec. 4.4).

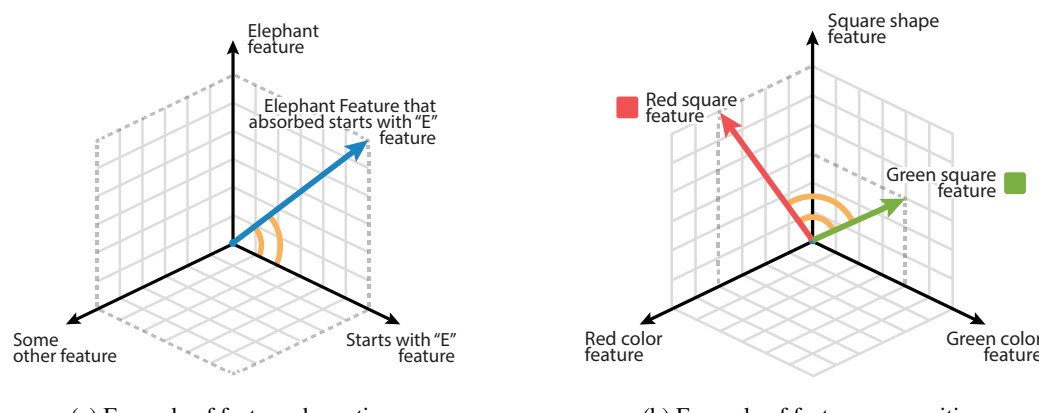

(a) Example of feature absorption

(b) Example of feature composition

Figure 2: **Illustration of feature absorption and feature composition**: (a) In feature absorption, specific features like "elephant" absorb broader features like "starts with E". (b) In feature composition, independent concepts like "red color" and "square form" are merged into composite features.

## 2 RELATED WORK

**SAEs for interpretability**   SAEs have gained attention as a tool for probing LLM internals, contributing to efforts to understand models that often function as "black boxes". A key issue is polysemanticity, where individual neurons respond to multiple unrelated concepts (Bricken et al., 2023). SAEs aim to resolve this by decomposing dense LLM activation vectors into a sparse set of monosemantic features, each representing a single concept (Cunningham et al., 2023; Huben et al., 2025). Pioneering work Bricken et al. (2023) and Cunningham et al. (2023) demonstrated the effectiveness of this approach on small transformers, finding interpretable features such as DNA sequences or legal text. Subsequent efforts scaled SAEs to larger models like Claude 3 Sonnet (Templeton et al., 2024) and GPT-4 (Gao et al., 2024), as well as open-source models (Lieberum et al., 2024).

**SAE architectures**   Limitations in basic SAEs, typically using ReLU activation with an L1 penalty (Bricken et al., 2023), such as L1-induced shrinkage (underestimation of feature strength) and difficulty in precise L0 control (Gao et al., 2024; Templeton et al., 2024) have driven architectural innovation. JumpReLU SAEs (Rajamanoharan et al., 2024), use a learned threshold within the activation function for direct L0 optimization. TopK SAE (Gao et al., 2024) selects only the top $K$ activations, simplifying tuning and reducing shrinkage compared to L1; BatchTopK SAE (Bussmann et al., 2024) further improves it by applying the TopK constraint at the batch level for adaptive sparsity and improved reconstruction. The latter approach appears promising for our purposes, as it allows precise sparsity control along with excellent reconstruction capabilities. For a detailed overview of the SAE variations, we further refer the reader to the survey by Shu et al. (2025).

**Challenges in SAE training**   Despite ongoing advancements, SAEs continue to face challenges: Chanin et al. (2024) describes the phenomenon of feature *absorption*, when broad features absorb into more specific ones. Leask et al. (2025) highlights feature *composition*, when independent features merge into one larger feature, and introduces the MetaSAE, which emerges as a promising approach to identify these problems.

**Hierarchical approaches**   Recently, Bussmann et al. (2025) proposed Matryoshka SAE to address these issues by employing a hierarchical approach. It builds upon BatchTopK architecture and uses nested features with increasing latent space size so that SAE separately learns broad and specific features. However, this hierarchical design leads to feature hedging (Chanin et al., 2025), where narrow higher-level dictionaries merge correlated features, reducing interpretability. Additionally, this approach introduces substantial computational overhead ($+50\%$ compared to traditional SAEs) and a degradation in reconstruction performance. Furthermore, while its reliance on hierarchical representation seems intuitive, its interpretability remains poorly explored.

**Orthogonal approaches** In contrast to hierarchical complexity, OrtSAE leverages orthogonality. This approach has been explored in deep learning to improve generalization and reduce overfitting (Wang et al., 2020; Rodríguez et al., 2016; Cogswell et al., 2015; Cha and Thiyagalingam, 2023). However, OrtSAE repurposes orthogonality to directly combat feature absorption and composition in SAEs. This approach naturally discourages feature merging while maintaining efficiency, avoiding Matryoshka's computational overhead and feature hedging while achieving comparable performance. Another line of work, Matching Pursuit SAE (MP-SAE) (Costa et al., 2025), enforces local orthogonality through recursive decomposition during forward pass. However, this design makes training 50-100× slower than traditional SAEs and may actually even encourage feature non-atomicity (see App N for detailed discussion). This limitation motivates our focus on global orthogonality constraints that maintain efficiency while directly addressing absorption and composition.

## 3 ORTHOGONAL SPARSE AUTOENCODERS

### 3.1 TRADITIONAL SPARSE AUTOENCODERS

Sparse autoencoders (SAEs) aim to reconstruct model activations $\mathbf{x} \in \mathbb{R}^n$ as a sparse linear combination of $m \gg n$ feature vectors, or *latents*. Formally, a SAE consists of an encoder and a decoder:

$$
\begin{aligned}
\mathbf{h}(\mathbf{x}) &= \sigma(\mathbf{W}^{\text{enc}}\mathbf{x} + \mathbf{b}^{\text{enc}}), \\
\hat{\mathbf{x}}(\mathbf{h}) &= \mathbf{W}^{\text{dec}}\mathbf{h} + \mathbf{b}^{\text{dec}}.
\end{aligned}
\tag{1}
$$

The encoder, followed by a non-linearity $\sigma(\cdot)$, learns a mapping from the activations to a sparse and overcomplete latent code $\mathbf{h}(\mathbf{x}) \in R^m$. Given $\mathbf{h}(\mathbf{x})$, the decoder reconstructs the original input as a sparse linear combination of latents, $\mathbf{W}_{\mathbf{i}}^{\text{dec}}, \mathbf{i} = 1, ..., m$, as $\hat{\mathbf{x}}$.

The standard loss function to train a SAE is defined as:

$$
\mathcal{L}(\mathbf{x}) = \underbrace{\|\mathbf{x} - \hat{\mathbf{x}}(\mathbf{h}(\mathbf{x}))\|_2^2}_{\mathcal{L}_{\text{reconstruct}}} + \underbrace{\lambda\, S(\mathbf{h}(\mathbf{x}))}_{\mathcal{L}_{\text{sparsity}}} + \alpha L_{\text{aux}},
\tag{2}
$$

where $S$ is a sparsity penalty and $\lambda$ is a coefficient controlling the trade-off between sparsity and reconstruction quality. The optional $L_{\text{aux}}$ term covers any auxiliary penalties (e.g. recycling dead units (Gao et al., 2024)).

Traditional SAEs focus on reducing reconstruction loss while increasing sparsity. The ReLU SAE (Bricken et al., 2023; Cunningham et al., 2023) uses the ReLU activation function and applies an L1 penalty to ensure sparsity in $\mathbf{h}(\mathbf{x})$. TopK SAE (Gao et al., 2024) achieves sparsity by zeroing all entries of $\mathbf{h}(\mathbf{x})$ except for the $K$ largest ones. BatchTopK (Bussmann et al., 2024) SAE further improves the idea by selecting the top $B \times K$ entries across a batch of $\mathbf{h}(\mathbf{x})$, allowing some examples to have more or less active latents.

### 3.2 CHALLENGES IN TRAINING SAEs

The standard SAE objective (Eq. 2) prioritizes both accurate reconstruction and sparsity. However, this sparsity pressure can lead to two key problems that undermine feature interpretability. First, in *feature absorption* (Fig. 2a), a seemingly monosemantic latent (e.g., for the concept "starts with E") develops arbitrary blind spots. For instance, it may fire on "echo" and "energy" but not on "elephant," because the SAE has found it more efficient to put the "starts with E" direction directly into a token-aligned "elephant" latent. This way, only one latent activates instead of two, satisfying the sparsity objective. The general feature is thus "absorbed" by the specific one in certain contexts, creating a misleading classifier. Second, *feature composition* (Fig. 2b) occurs when the SAE conflates distinct concepts that often appear together (like "red" and "square") into a single, polysemantic latent for "red square," failing to learn the underlying atomic features.

Feature absorption and composition produce redundant representations where multiple latents capture overlapping concepts, leading to high cosine similarities between decoder vectors. Formally, consider two atomic features A ("red") and B ("square") (Fig. 2b). In traditional SAEs, these independent features can merge into a composite feature C ("red square"). Let $\mathbf{W}_A^{\text{dec}}, \mathbf{W}_B^{\text{dec}}$, and $\mathbf{W}_C^{\text{dec}}$ denote the

decoder vectors for features A, B, and C, respectively. When feature composition occurs, C incorporates components of both features A and B. This creates higher correlations between C and each atomic feature: $\cos(\mathbf{W}_C^{\text{dec}}, \mathbf{W}_A^{\text{dec}}) > \cos(\mathbf{W}_A^{\text{dec}}, \mathbf{W}_B^{\text{dec}})$ and $\cos(\mathbf{W}_C^{\text{dec}}, \mathbf{W}_B^{\text{dec}}) > \cos(\mathbf{W}_A^{\text{dec}}, \mathbf{W}_B^{\text{dec}})$ (for full formal derivation see App. K). Similarly, feature absorption creates overlapping latents, resulting in the decoder vectors that are more correlated than they should be for truly atomic features. To address these issues, OrtSAE extends the traditional SAE objective by enforcing orthogonality between SAE latents. This geometric constraint directly prevents features from capturing overlapping concepts, tackling absorption and composition at their root.

Our approach assumes atomic features should be nearly orthogonal. While some concepts (e.g., days of the week or months of the year) naturally form non-orthogonal circular structures (Engels et al., 2024), these appear to be exceptional cases, as the study found no other readily interpretable multidimensional features. If non-orthogonal geometries were widespread, our penalty would force an incorrect basis and degrade reconstruction. However, OrtSAE maintains high reconstruction fidelity (Sec. 4.2), confirming orthogonality is valid for most features. Furthermore, our soft penalty described below (Sec. 3.3) discourages but does not prevent non-orthogonal representations, allowing the model to learn necessary exceptions when needed.

### 3.3 OrtSAE Training Procedure

The main contribution of OrtSAE is the introduction of a new orthogonalization penalty that penalizes high similarities between SAE latents. Formally, given a SAE with decoder matrix $\mathbf{W}^{\text{dec}} \in \mathbb{R}^{n \times m}$, we first define the cosine similarity between two feature vectors as:

$$\cos(\mathbf{W}_{\mathbf{i}}^{\text{dec}}, \mathbf{W}_{\mathbf{j}}^{\text{dec}}) = \frac{\langle \mathbf{W}_{\mathbf{i}}^{\text{dec}}, \mathbf{W}_{\mathbf{j}}^{\text{dec}} \rangle}{\max(\|\mathbf{W}_{\mathbf{i}}^{\text{dec}}\|_2 \cdot \|\mathbf{W}_{\mathbf{j}}^{\text{dec}}\|_2, \delta)}, \tag{3}$$

where $\delta > 0$ is a small constant added to prevent division by zero. Using this definition, we formulate our orthogonality penalty as:

$$L_{\text{orthogonal}}(\mathbf{W}^{\text{dec}}) = \frac{1}{K(m)} \sum_{k=1}^{K(m)} \frac{1}{|C_k|} \sum_{i \in C_k} \left( \max_{\substack{j \in C_k \\ j \neq i}} \cos(\mathbf{W}_{\mathbf{i}}^{\text{dec}}, \mathbf{W}_{\mathbf{j}}^{\text{dec}}) \right)^2. \tag{4}$$

Instead of computing all pairwise similarities between feature vectors $\mathbf{W}_{\mathbf{i}}^{\text{dec}}$, which would require $\mathcal{O}(m^2)$ operations and is infeasible for large $m$, at each training step we *randomly* partition the latent space into $K := K(m)$ equal chunks, each containing a fixed number of latents, $|C_k|, k = 1, ..., K(m)$, proportional to $m$. Within each $k$-th chunk, we find the maximum pairwise cosine similarity between every $\mathbf{W}_{\mathbf{i}}^{\text{dec}}, \mathbf{i} \in C_k$ and all other latents from $C_k$, square this value to penalize highly correlated features more and compute the expectation. We compute the final value by averaging across all $K$ chunks. This chunk-wise strategy reduces the computational complexity to $\mathcal{O}(m)$ and provides an efficient scaling strategy to a larger latent spaces.

With the orthogonal penalty defined, OrtSAE training objective is defined as:

$$\mathcal{L}_{\text{OrtSAE}}(x) = L_{\text{MSE}} + \lambda L_{\text{sparsity}} + \alpha L_{\text{aux}} + \gamma L_{\text{orthogonal}}, \tag{5}$$

where $\gamma$ is an *orthogonality coefficient* that controls the strength of the applied penalty.

To further enhance computational efficiency, we explore computing the orthogonality loss every fifth training iteration, scaling the orthogonality coefficient $\gamma$ by a factor of 5 to maintain regularization strength, which yields comparable performance with significantly reduced computational overhead, as detailed in App. C.

## 4 Experiments

Here, we present our experimental evaluation of OrtSAE. Sec. 4.1 covers the experimental setup. Sec. 4.2 compares OrtSAE's core performance metrics to other methods. Sec. 4.3 presents quantitative and qualitative results on feature atomicity. Sec. 4.4 assesses OrtSAE's performance on downstream tasks using SAEBench.

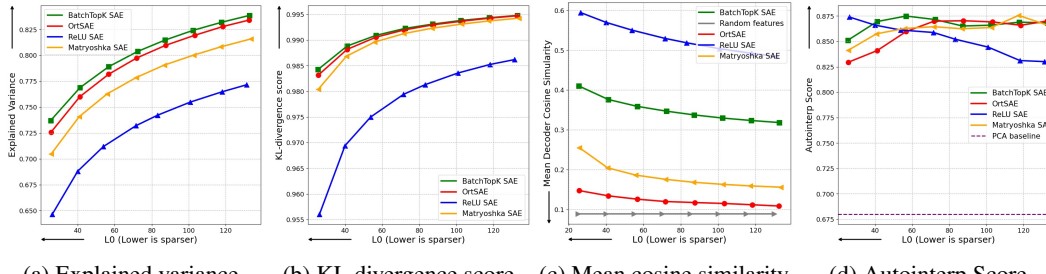

(a) Explained variance.    (b) KL divergence score.    (c) Mean cosine similarity.    (d) Autointerp Score.

Figure 3: **Core performance metrics.** (a) Explained variance: OrtSAE shows slightly lower reconstruction fidelity than BatchTopK SAE but outperforms Matryoshka SAE. (b) KL-divergence score: OrtSAE matches BatchTopK SAE and exceeds Matryoshka SAE. (c) Mean cosine similarity to closest decoder feature: OrtSAE achieves near-random initialization orthogonality, significantly lower than other SAE variants. (d) Autointerp Score: OrtSAE demonstrates interpretability comparable to both BatchTopK and Matryoshka SAEs.

## 4.1 EXPERIMENTAL SETUP.

**Baselines.** We compare our model with: 1) traditional SAEs, such as ReLU SAE (Bricken et al., 2023; Cunningham et al., 2023) and state-of-the-art BatchTopK SAE (Bussmann et al., 2024); 2) recent Matryoshka SAE that enforce nested, hierarchical learning at multiple feature levels (Bussmann et al., 2025).

**Models Configuration.** Following the work of Bussmann et al. (2025) we train SAEs on the activations from layer 12 of the Gemma-2-2B (26 layers total). Each SAE has latent space of size $m = 65536$ and sparsity levels L0 in $\{25, 40, 55, 70, 85, 100, 115, 130\}$. The training uses 500 million tokens from the OpenWebText dataset (Gokaslan and Cohen, 2019) with a context length of 1024. As a basis of OrtSAE, we follow BatchTopK SAE repository, leveraging BatchTopK's precise L0 sparsity control. For OrtSAE, we set the number of chunks $K(m) = \lceil m/8192 \rceil$ (yielding $K = 8$ for $m = 65536$) with $\gamma = 0.25$. The full details and hyperparameters are available in the App. A. To assess the transferability of our approach, we also conduct experiments on layer 20 of Gemma-2-2B and layer 20 of Llama-3-8B and report the results in App. B.To ensure the reproducibility, we will publicly release all code and hyperparameters.

**Evaluation.** Following Gao et al. (2024) and Bussmann et al. (2025), we evaluate SAEs core performance through explained variance, KL-divergence, orthogonality, and feature interpretability. Atomicity analysis includes absorption metrics, MetaSAE-based composition rate (Leask et al., 2025), clustering, cross-model overlap, and qualitative investigation. Downstream assessment uses SAEBench (Karvonen et al., 2025) with spurious correlation removal, targeted probe perturbation, sparse probing, and RAVEL tasks.

## 4.2 FOUNDATIONAL PERFORMANCE ANALYSIS

We evaluate OrtSAE using four metrics: (1) reconstruction fidelity, computed as the fraction of explained variance; (2) downstream predictive performance, measured by the KL-divergence score between the original LLM's output distributions and those generated using reconstructed activations; (3) feature orthogonality, calculated as the mean cosine similarity to each feature's nearest decoder neighbor; and (4) feature interpretability, assessed via the Autointerp Score. Fig. 3 shows these metrics across sparsity levels, demonstrating OrtSAE's balance between reconstruction quality and model functionality preservation. To further validate the generalizability of our findings across different model architectures and layers, we conducted additional experiments on layer 20 of Gemma-2-2B and layer 20 of Llama-3-8B, with results reported in App. B. We also analyze the impact of varying the number of chunks on OrtSAE performance to assess its robustness and scalability, with details provided in App. C.

**Reconstruction and Predictive Performance.** We first evaluate reconstruction quality through explained variance, measuring how accurately each SAE reconstructs input activations. To assess whether these reconstructions preserve the model's functionality, we additionally examine downstream predictive performance using KL-divergence scores (detailed descriptions provided in App.D). We also tested the effect of decoded activations on the base language model's perplexity using LogLoss, which shows the same patterns as the KL-divergence scores (see App. E). Fig. 3a shows OrtSAE achieves comparable performance to BatchTopK SAE while outperforming Matryoshka SAE by 2% in fraction of explained variance. The KL-divergence scores (Fig. 3b) reveal nearly identical predictive behavior between OrtSAE and BatchTopK SAE, with both slightly surpassing Matryoshka SAE. These results are particularly notable because OrtSAE maintains strong performance despite its additional orthogonality constraints, whose effects we analyze next through feature similarity.

**Feature Orthogonality.** To quantify the separation of decoder features, we compute the mean cosine similarity (MeanCosSim) for each feature vector $i$ to its closest neighbor in the decoder matrix:

$$\text{MeanCosSim} = \frac{1}{m} \sum_{i=1}^{m} \max_{j \neq i} \cos(\mathbf{W}_i^{\text{dec}}, \mathbf{W}_j^{\text{dec}}). \tag{6}$$

As shown in Fig. 3c, OrtSAE achieves superior separation with MeanCosSim values 2.7 times lower than BatchTopK and 1.5 times lower than Matryoshka SAE, approaching random initialization levels. This enhanced orthogonality directly contributes to improved feature atomicity, as shown in Sec. 4.3. We further validate this reduced feature interference using Babel scores in App. M, which confirm OrtSAE's superior dictionary coherence.

**Feature Interpretability.** The Autointerp Score evaluates feature interpretability using GPT-4o-mini as an LLM judge. Following Paulo et al. (2024)'s methodology, we first generate interpretable descriptions of 1,000 latents using an LLM, then quantitatively assess these explanations by measuring how accurately the LLM can predict whether each latent activates (fires) on new input tokens (detailed descriptions provided in Appx. F). Fig. 3d demonstrates comparable interpretability between OrtSAE, BatchTopK SAE, and Matryoshka SAE latents across all tested sparsity levels.

These results demonstrate that OrtSAE achieves an optimal balance - matching top reconstruction performance while significantly improving feature separation without compromising interpretability. This demonstrates that our new training procedure can enhance feature quality while preserving model functionality, as we explore further in our atomicity analysis.

### 4.3 ATOMICITY ANALYSIS

We assess feature atomicity through four quantitative measures: (1) MetaSAE-based composition rates, (2) absorption metrics, (3) clustering coefficients, and (4) cross-model feature uniqueness, complemented by qualitative analysis of decomposed features. Fig. 4 presents the quantitative comparisons across sparsity levels, while Fig. 5 illustrates OrtSAE's ability to disentangle composite features through concrete examples. Additionally we also validate OrtSAE's superior atomic feature recovery on synthetic data with known ground truth in App. J.

**MetaSAE-Based Feature Composition Analysis.** We train a MetaSAE on the decoder features of SAEs, following the methodology of Leask et al. (2025). The MetaSAE follows the same training procedure as ordinary SAEs (Sec. 4.1) but operates on decoder features rather than LLM activations, attempting to decompose these higher-level representations into more atomic latent components (detailed descriptions provided in Appx. G). The MetaSAE employs a BatchTopK architecture with sparsity $k = 4$ and a dictionary size reduced to 25% of the original SAEs. We measure composition via explained variance, where lower values indicate higher atomicity. Fig. 4a shows that OrtSAE attains a composition rate much lower (by 0.06 at L0 of 70) than BatchTopK SAE and similar to Matryoshka SAE, reflecting pronounced feature atomicity and resistance to decomposition into simpler constituents.

**Feature Absorption Analysis.** Feature absorption is evaluated using SAEBench (Karvonen et al., 2025). Following established methodologies for studying absorption in sparse autoencoders (Chanin et al., 2024), we perform tests in the first-letter classification and hierarchical concept domains

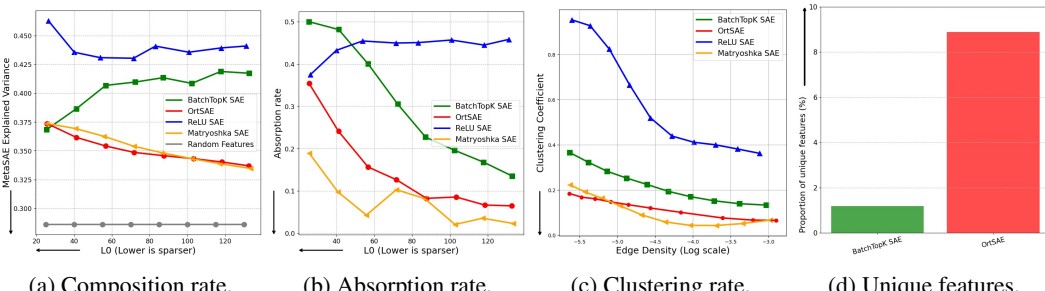

(a) Composition rate.     (b) Absorption rate.     (c) Clustering rate.     (d) Unique features.

Figure 4: **Atomicity metrics.** (a) MetaSAE-based Composition rate: OrtSAE shows reduced feature merging compared to traditional SAEs and matches Matryoshka SAE. (b) Absorption rate: OrtSAE shows significant improvement over BatchTopK with performance approaching Matryoshka SAE. (c) Feature clustering: OrtSAE matches Matryoshka SAE in low feature interconnection, both outperforming traditional SAEs. (d) Proportion of unique features: OrtSAE discovers substantially more distinct features than BatchTopK SAE.

(detailed descriptions provided in Appx. B). Fig. 4b shows that OrtSAE achieves a significantly reduced absorption rate compared to BatchTopK SAE (by 0.17 at L0 of 70), but slightly higher than Matryoshka SAE, indicating effective minimization of conceptual overlap compared to traditional SAEs.

**Clustering Properties of Decoder Features.** The clustering coefficient measures how tightly connected features are in a graph of decoder interactions, where edges represent high cosine similarity between features (detailed descriptions provided in Appx. C). A lower coefficient indicates that features are more independent, forming fewer interconnected groups, which suggests greater feature atomicity. We evaluate this across 10 similarity thresholds at varying edge densities (the ratio of existing edges to the number of all possible edges). Fig. 4c shows the clustering coefficient of OrtSAE is substantially lower than that of BatchTopK SAE and similar to Matryoshka SAE, signifying enhanced feature independence compared to traditional SAEs.

**Cross-Model Feature Overlap Analysis.** We measure feature uniqueness by computing maximum pairwise cosine similarity between OrtSAE and BatchTopK SAE features, at a L0 of 70. A feature is considered unique if all cross-model similarities are below 0.2 (see App. L for threshold sensitivity analysis). OrtSAE retains 9% unique features, compared to 1.5% for BatchTopK (Fig. 4d). This six-fold increase in unique features highlights OrtSAE's ability to discover novel features, enhancing scalability for larger dictionaries.

**Qualitative Analysis of Feature Atomicity.** We analyze feature atomicity by decomposing Batch-TopK SAE features into sparse combinations of OrtSAE features, following a methodology adapted from MetaSAE (Leask et al., 2025). We select BatchTopK SAE (at L0 of 70) features if their OrtSAE (at L0 of 70) approximation has a cosine similarity above 0.95 and each coefficient is at least 0.1, ensuring meaningful contributions. Fig. 5 illustrates the decomposition of a BatchTopK SAE feature for "Queen terms" into two OrtSAE features: one for "Queen terms" and another for "titles and royal concepts". This demonstrates how OrtSAE disentangles broader concepts absorbed by specialized features. Additional examples of feature decompositions are provided in Appx. D.

These results demonstrate OrtSAE improves feature atomicity over traditional SAEs, achieving lower composition and absorption rates, reduced clustering, and a higher proportion of unique features. Qualitative evidence supports these findings, demonstrating the effective decomposition of complex BatchTopK features by OrtSAE features. These results highlight the efficacy of orthogonality constraint in yielding disentangled representations, paving the way for downstream tasks.

## 4.4 DOWNSTREAM BENCHMARKS

We evaluate OrtSAE using the SAEBench (Karvonen et al., 2025), which measures SAEs quality across a diverse set of tasks related to practical downstream applications. The key metrics we report

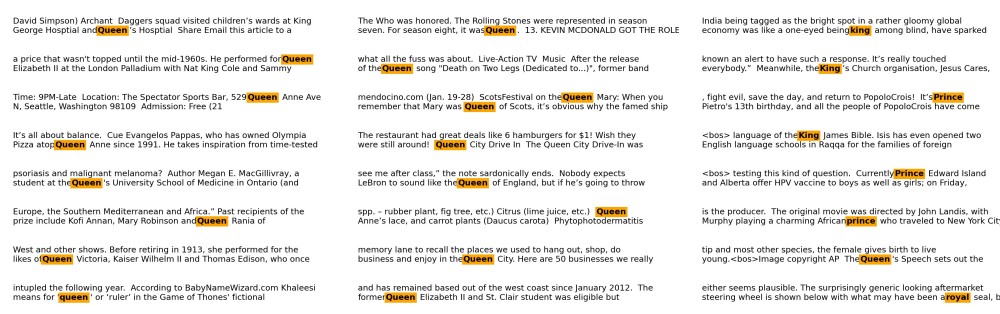

(a) BatchTopK SAE feature that activates only on "Queen" token

(b) OrtSAE feature that activates only on "Queen" token

(c) OrtSAE feature that activates only on titles and royal concepts

Figure 5: **Decomposition of a BatchTopK SAE Feature into OrtSAE Features.** (a) A BatchTopK SAE feature activating solely on the token "Queen", expressed as a linear combination of two OrtSAE features: (b) An OrtSAE feature specific to the token "Queen". (c) An OrtSAE feature capturing royal titles, revealing how OrtSAE disentangles the broader royalty concept absorbed by the "Queen" feature in BatchTopK SAE.

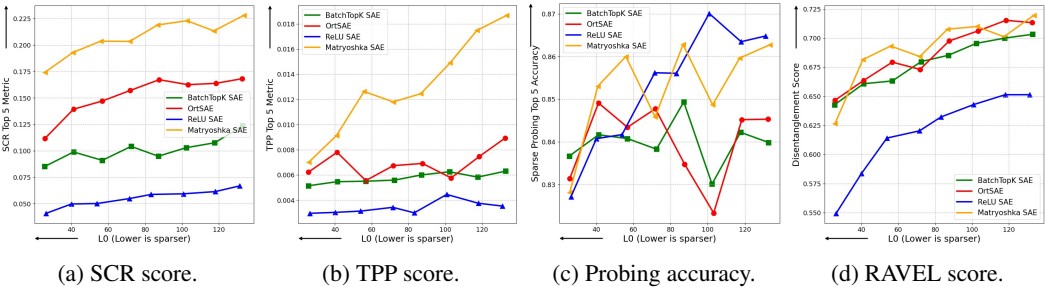

(a) SCR score.  (b) TPP score.  (c) Probing accuracy.  (d) RAVEL score.

Figure 6: **Results on SAEBench.** (a) Spurious Correlation Removal: OrtSAE outperforms traditional SAEs but achieves lower scores than Matryoshka. (b) Targeted Probe Perturbation: OrtSAE shows modest improvements over traditional SAEs, while Matryoshka demonstrates the strongest performance. (c) Sparse Probing: OrtSAE maintains accuracy comparable to baseline methods. (d) RAVEL: OrtSAE achieves scores similar to both BatchTopK and Matryoshka SAEs.

on are Spurious Correlation Removal (SCR), Targeted Probe Perturbation (TPP), Sparse Probing, and RAVEL.

**Spurious Correlation Removal and Targeted Probe Perturbation.** SCR evaluates the removal of spurious correlations (e.g., gender in profession classification) by zero-ablating SAE latents, while TPP tests class-specific concept isolation in multi-class settings via targeted ablation. Fig. 6 demonstrates that OrtSAE achieves stronger performance than traditional SAEs in both tasks, with a significant improvement in SCR scores and modest gains in TPP compared to BatchTopK SAE. While Matryoshka SAE achieves the highest absolute scores, OrtSAE delivers similarly strong results with lower computational overhead and traditional architecture.

**Sparse Probing.** Sparse Probing evaluates the ability of SAEs to isolate specific concepts, such as sentiment, within individual latents without explicit supervision. For each concept, we select the top-$k$ latents by comparing their mean activations on positive versus negative examples. A linear probe is then trained on these latents to predict the concept. High probe accuracy indicates that the latents effectively capture the target concept in a disentangled manner. As shown in Fig. 6c, OrtSAE achieves probing accuracy comparable to BatchTopK and Matryoshka SAEs across various sparsity levels, demonstrating its capability to produce interpretable, concept-aligned features.

**RAVEL.** RAVEL (Resolving Attribute–Value Entanglements in Language Models) (Huang et al., 2024) evaluates disentanglement by manipulating attributes in LLM activations (e.g., transferring city-related features from "Tokyo" to "Paris") while minimizing interference with unrelated attributes (e.g., France-related context). Fig. 6d demonstrates that OrtSAE, BatchTopK SAE, and Matryoshka SAE exhibit equivalent performance, highlighting OrtSAE's ability to enable precise interventions without compromising efficacy.

The different results across tasks can be explained by what each task requires. OrtSAE's orthogonality penalty reduces feature overlaps. This is very important for Spurious Correlation Removal (SCR), where the goal is to remove one feature without affecting others. This leads to a big performance gain. For Targeted Probe Perturbation (TPP), which deals with multiple classes, the benefit is smaller because some feature relationships are less problematic. For Sparse Probing and RAVEL, which test how well features capture single concepts or can be transferred, OrtSAE's orthogonal features work just as well as others, giving similar performance. Thus, orthogonality most benefits tasks vulnerable to feature entanglement, underscoring OrtSAE's ability to produce atomic features without compromising utility.

## 5 CONCLUSION

In this work, we introduce OrtSAE, a novel sparse autoencoder training approach that enhances latent atomicity through orthogonal constraints on decoder features. This method effectively addresses feature absorption and composition, key obstacles to interpretable representations, while preserving reconstruction fidelity comparable to traditional SAEs. Notably, OrtSAE achieves this with minimal computational overhead through an efficient chunk-wise orthogonalization penalty that scales linearly with feature count. Experiments across different language model families demonstrate OrtSAE's significant reductions in absorption and composition rates, yielding 9% more distinct features, superior spurious correlation removal (+6%), and on-par performance across other SAEBench tasks. These insights underscore geometric constraints' role in disentangling superposed representations, offering fresh perspectives on the superposition hypothesis. Future work should investigate using orthogonal features as more interpretable building blocks for neural circuit discovery, potentially leading to clearer mechanistic models of model computations.

## 6 REPRODUCIBILITY STATEMENT

To ensure the reproducibility of our work, we have taken several measures throughout the paper and supplementary materials. All experimental details, including model architectures, hyperparameters, and training procedures, are comprehensively documented in Section 4.1 and App. A. We provide complete specifications for our OrtSAE implementation, including the orthogonal penalty formulation and chunking strategy in Section 3.3. The evaluation metrics and benchmarks are described in detail in Sections 4.2-4.4, with additional methodological explanations in Appendices B-G. Code for OrtSAE training, evaluation scripts, feature analysis tools, and SAEBench integration will be released anonymously as supplementary material and made fully public upon acceptance. The datasets used in our experiments (OpenWebText) are publicly available, and we specify exact data processing steps in App. A. For the SAEBench evaluations, we follow established protocols from prior work with detailed descriptions of each task. Additional experiments on different model architectures and layers (App. B) and ablation studies on key hyperparameters (App. C) further validate the robustness of our approach.

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

## A  ADDITIONAL DETAILS OF SAE TRAINING SETUP

Following the approach of (Bussmann et al., 2025), we train Sparse Autoencoders (SAEs) on activations from layer 12 of the Gemma-2-2B model (Team et al., 2024) (26 layers total) to align with prior work. To assess the generalizability of our approach, we also conduct experiments on layer 20 of Gemma-2-2B and layer 20 of Llama-3-8B (Dubey et al., 2024) (32 layers total). Each SAE has latent space of size $m = 65536$ and sparsity levels L0 in $\{25, 40, 55, 70, 85, 100, 115, 130\}$. The training uses 500 million tokens from the OpenWebText dataset (Gokaslan and Cohen, 2019) with a context length of 1024. We employ the AdamW optimizer (Loshchilov and Hutter, 2017) with a learning rate of $2 \times 10^{-4}$ and a batch size of 2048. All SAE variants except ReLU SAE employ an auxiliary loss coefficient $\alpha = 1/32$ to mitigate dead features during training.

As a basis of OrtSAE, we follow BatchTopK SAE repository, leveraging BatchTopK's precise L0 sparsity control. For OrtSAE, we set the number of chunks $K(m) = \lceil m/8192 \rceil$ (yielding $K = 8$ for $m = 65536$) with $\gamma = 0.25$. This orthogonality coefficient was selected through hyperparameter sweep over [0.1, 0.5], representing the smallest value that reliably achieves near-random initialization orthogonality while maintaining reconstruction performance. This value has proven stable across different models and layers. For Matryoshka SAE, we use the group sizes recommended in the original paper: $\{0.03125, 0.0625, 0.125, 0.25, 0.53125\}$.

All SAEs are trained comparably with identical data ordering and hyperparameters on 1 H100 GPU, 80GB. For one SAE, training requires approximately 10 hours. To ensure reproducibility, we will publicly release all code, hyperparameters, and instructions for accessing the datasets. The results of additional experiments on layer 20 of Gemma-2-2B and Llama-3-8B focusing on a sparsity level of L0 = 70 are reported in the App. B.

## B  ADDITIONAL EXPERIMENTS ON GEMMA-2-2B AND LLAMA-3-8B

To address the generalizability of our findings, we conducted additional experiments on layer 20 of Gemma-2-2B (26 layers total) and layer 20 of Llama-3-8B (32 layers total) (Dubey et al., 2024), following the experimental setup described in Sec. 4.1. We trained SAEs with a sparsity level of L0 = 70 and measured key metrics: explained variance, mean cosine similarity, composition rate, absorption rate, and Spurious Correlation Removal (SCR) score. The results, presented in Tables 1 and 2, confirm the findings from layer 12 of Gemma-2-2B, showing consistent reductions in feature absorption and composition, as well as improved performance in tasks such as Spurious Correlation Removal. Notably, the Explained Variance gap between Matryoshka SAE and OrtSAE widens to 0.04 in Llama-3-8B (0.722 vs. 0.762), reinforcing OrtSAE's advantage in reconstruction performance.

Table 1: **Performance of SAEs trained on layer 20 of Gemma-2-2B with L0=70.**

| SAE model | Expl. var. | Mean Cos. sim. | Comp. rate | Abs. rate | SCR score |
|---|---|---|---|---|---|
| ReLU SAE | 0.784 | 0.549 | 0.527 | 0.371 | 0.144 |
| BatchTopK SAE | **0.843** | 0.354 | 0.490 | 0.220 | 0.308 |
| Matryoshka SAE | 0.811 | 0.148 | 0.349 | **0.015** | **0.385** |
| OrtSAE | 0.836 | **0.112** | **0.340** | 0.095 | 0.322 |

Table 2: **Performance of SAEs trained on layer 20 of Llama-3-8B with L0=70.**

| SAE model | Expl. var. | Mean Cos. sim. | Comp. rate | Abs. rate | SCR score |
|---|---|---|---|---|---|
| ReLU SAE | 0.704 | 0.517 | 0.413 | 0.490 | 0.060 |
| BatchTopK SAE | **0.769** | 0.327 | 0.461 | 0.148 | 0.103 |
| Matryoshka SAE | 0.722 | 0.149 | 0.323 | **0.022** | **0.191** |
| OrtSAE | 0.762 | **0.107** | **0.316** | 0.070 | 0.151 |

## C    EFFECT OF NUMBER OF CHUNKS ON ORTSAE PERFORMANCE

We evaluated the impact of the number of chunks and the frequency of orthogonality loss computation on OrtSAE performance, following the experimental setup in Sec. 4.1. OrtSAE was tested with chunk counts $K \in \{4, 8, 16, 32, 64\}$, using a fixed sparsity at L0 of 70. Additionally, to enhance computational efficiency, we explored a modified OrtSAE variant where the orthogonality loss is computed every fifth training iteration, with the orthogonality coefficient $\gamma$ scaled by a factor of 5 to maintain regularization strength.

As shown in Figure 7, OrtSAE demonstrates robust performance in core reconstruction and atomicity metrics across different numbers of chunks. The original OrtSAE and the modified OrtSAE show similar trends, with the modified version maintaining performance even better then original. Explained variance (Fig.7a) remains stable around 0.77–0.78 across chunk counts for both variants, slightly outperforming ReLU SAE and Matryoshka SAE. Mean cosine similarity (Fig.7b) increases modestly with more chunks (e.g., from 0.10 at $K = 4$ to 0.20 at $K = 64$), but remains lower than BatchTopK and ReLU SAEs. Absorption rate (Fig.7c) and composition rate (Fig.7d) also increase slightly with more chunks (e.g., absorption from 0.15 to 0.20), yet both variants outperform traditional SAEs. The modified OrtSAE achieves performance within 1–2% of the original OrtSAE across these metrics, reducing the computational overhead of the orthogonality loss by approximately five times, as it is calculated in only 20% of training iterations. Overall, OrtSAE demonstrates both robustness and scalability by maintaining core SAE performance across varying chunk counts while reducing training overhead to within 4% of the BatchTopK baseline (see Table 3).

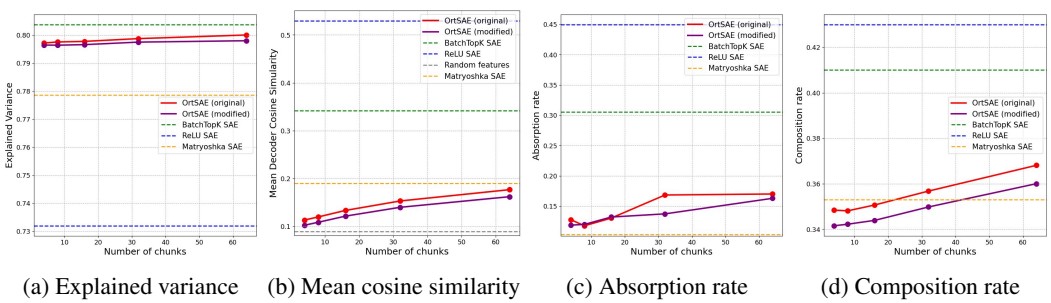

(a) Explained variance    (b) Mean cosine similarity    (c) Absorption rate    (d) Composition rate

Figure 7: **Performance of OrtSAE across different number of chunks at L0 of 70**. OrtSAE shows robust performance in core reconstruction and atomicity metrics across different number of chunks.

Table 3: **Training times for SAE variants.** Ratios are relative to BatchTopK SAE.

| Method | Training Time (minutes) | Time Ratio* |
|---|---|---|
| BatchTopK SAE | 325 | 1.0× |
| Matryoshka SAE | 373 | 1.15× |
| OrtSAE modified ($K = 8$) | 361 | 1.11× |
| OrtSAE modified ($K = 64$) | **340** | **1.04×** |

*Relative to BatchTopK SAE

## D    KL-DIVERGENCE SCORE DEFINITION

The KL-divergence score assesses how effectively a sparse autoencoder (SAE) preserves the predictive behavior of a language model by comparing next-token probability distributions. We define $P_{\text{orig}}$ as the distribution from the original model, $P_{\text{SAE}}$ as the distribution when activations are replaced by SAE reconstructions, and $P_{\text{ablated}}$ as the distribution when activations are set to zero, serving as a baseline.

The score is given by:

$$\text{KL-Divergence Score} = \frac{D_{KL}(P_{\text{ablated}} \parallel P_{\text{orig}}) - D_{KL}(P_{\text{SAE}} \parallel P_{\text{orig}})}{D_{KL}(P_{\text{ablated}} \parallel P_{\text{orig}})}$$

where $D_{KL}$ denotes the Kullback-Leibler divergence. This metric ranges from 0, indicating no improvement over the zero-ablated baseline, to 1, indicating perfect reconstruction ($P_{\text{SAE}} = P_{\text{orig}}$). In SAEBench, the score is averaged over a dataset to evaluate SAE reconstruction quality.

## E LogLoss Results

To further evaluate the impact of decoded activations on the base language model's predictive performance, we compute LogLoss scores for SAEs trained on layer 12 of Gemma-2-2B across various sparsity levels (L0 $\in \{40, 70, 100, 130\}$). LogLoss measures the negative log-likelihood of the model's next-token predictions, with lower values indicating better preservation of the base model's predictive behavior. The results, shown in Table 4, align closely with the KL-divergence findings (App. D), confirming that OrtSAE maintains predictive performance comparable to BatchTopK SAE, with both outperforming ReLU SAE and Matryoshka SAE.

Table 4: **LogLoss of SAEs trained on layer 12 of Gemma-2-2B.** Lower LogLoss indicates better preservation of the base language model's predictive performance.

| SAE model | L0=40 | L0=70 | L0=100 | L0=130 |
|---|---|---|---|---|
| No SAE (LLM's LogLoss) | 2.4533 | 2.4533 | 2.4533 | 2.4533 |
| ReLU SAE | 2.7657 | 2.6562 | 2.6094 | 2.5935 |
| BatchTopK SAE | **2.5623** | **2.5312** | **2.5152** | **2.5020** |
| Matryoshka SAE | 2.5786 | 2.5321 | 2.5161 | 2.5025 |
| OrtSAE | 2.5646 | 2.5319 | 2.5159 | 2.5024 |

## F Feature Interpretability Metrics Details

To evaluate the interpretability of Sparse Autoencoder (SAE) features, we follow the automated interpretability methodology outlined in (Paulo et al., 2024; Karvonen et al., 2025), leveraging large language models (LLMs) to generate and validate human-readable feature descriptions. We select 1,000 random SAE features, excluding "dead" features. For each feature, we collect up to 10 top-activating sequences from the OpenWebText dataset (Gokaslan and Cohen, 2019) and prompt GPT-4o-mini to generate a concise description, such as "sentiment terms" or "math expressions," capturing the feature's core concept.

To assess these descriptions, we create a test set for each feature with 100 sequences: 50 activating the feature at varying strengths and 50 random non-activating sequences, all sourced from OpenWebText. A separate GPT-4o-mini model predicts whether each sequence activates the feature based on the description, treating it as a binary (yes/no) classification task. The *Autointerp Score*, shown in Fig. 3d, is the prediction accuracy, measuring how well the description generalizes to new data. A high score indicates a monosemantic, interpretable feature, while a lower score may suggest polysemanticity or an inaccurate description. OrtSAE demonstrates interpretability comparable to BatchTopK and Matryoshka SAEs across sparsity levels (Sec. 4.2), confirming its ability to produce clear, disentangled feature representations.

## G MetaSAE-Based Feature Composition Metrics Details

The MetaSAE-based feature composition analysis, originally proposed by Leask et al. (2025) and extended by Bussmann et al. (2025), provides a quantitative method for assessing the atomicity of features learned by sparse autoencoders. This approach measures the degree of feature composition by training a secondary sparse autoencoder (MetaSAE) on the feature vectors of the primary SAE to decompose them into more atomic components.

In our implementation, we train MetaSAEs on the decoder weight matrices of both OrtSAE and BatchTopK SAE. The decoder weights are treated as input data points, where each feature vector from the primary SAE's dictionary serves as an input to the MetaSAE. The MetaSAE follows the same BatchTopK architecture as our primary SAEs but operates on this different input space.

The MetaSAE is configured with a dictionary size equal to one-quarter of the primary SAE's dictionary size (16,384 meta-latents for our primary SAEs with 65,536 features). We apply a sparsity constraint ensuring an average of 4 active meta-latents per decoder vector reconstruction. The MetaSAE learns to reconstruct each primary SAE feature vector using a sparse combination of meta-latents, where the meta-features represent atomic sub-components of the primary SAE's features.

The key metric for assessing composition is the explained variance, which measures the proportion of variance in the primary SAE's decoder weights that the MetaSAE can reconstruct. A higher explained variance indicates that the MetaSAE can effectively reconstruct the primary SAE's features using shared, atomic sub-features, suggesting that the original features were composed of these simpler components. Conversely, a lower explained variance implies that the primary features are already atomic and resist decomposition into simpler constituents. In our experiments, we interpret lower MetaSAE explained variance as indicating better feature atomicity in the primary SAE.

This methodology provides an objective, quantitative measure of feature composition that complements our qualitative analyses and other atomicity metrics, offering insights into the hierarchical structure of the representations learned by different SAE variants.

## H    CLUSTERING COEFFICIENT DEFINITION

The global clustering coefficient quantifies the tendency of nodes in a graph to form clusters. For Sparse Autoencoders (SAEs), nodes represent decoder features, and edges connect feature pairs with cosine similarity above a threshold.

Edge density, the proportion of possible edges present, is defined as:

$$\text{density} = \frac{2E}{n(n-1)},$$

where $E$ is the number of edges and $n$ is the number of nodes. Varying the similarity threshold adjusts the density, enabling analysis across connectivity levels.

The clustering coefficient $C$ is computed as:

$$C = \frac{3 \times \text{number of triangles}}{\text{number of connected triples}},$$

where a triangle is three nodes fully connected by edges, and a connected triple is three nodes linked by at least two edges. $C$ ranges from 0 (no clustering) to 1 (maximal clustering).

## I    ADDITIONAL QUALITATIVE EXAMPLES

We provide three additional examples of BatchTopK SAE features decomposed into orthogonal, atomic OrtSAE components. These cases further illustrate how OrtSAE's orthogonality constraint systematically disentangles composite concepts by breaking down complex features into their fundamental building blocks. The OrtSAE decompositions reveal how broader conceptual categories are separated into distinct, specialized features, demonstrating the method's ability to identify and preserve atomic feature representations that traditional SAEs often merge together. We also show BatchTopK SAE features decomposed into Matryoshka SAE features for comparison, illustrating how different approaches handle feature decomposition.

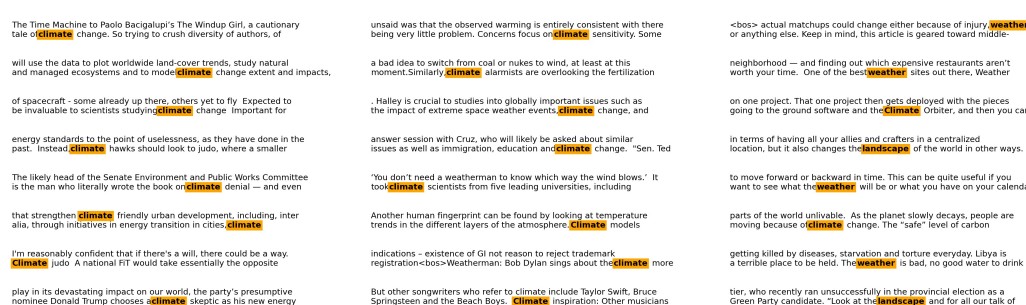

(a) BatchTopK SAE feature activating on climate-related terms.

(b) OrtSAE feature activating on on climate-related terms.

(c) OrtSAE feature activating on broader environmental and weather contexts.

Figure 8: **Decomposition of Climate-Related BatchTopK SAE Feature into OrtSAE Features.** (a) A BatchTopK SAE (L0=70) feature that activates on climate-related terms, which can be represented as a linear combination of two OrtSAE (L0=70): (b) An OrtSAE feature that activates specifically on climate-related terms. (c) An OrtSAE feature that activates on broader environmental contexts.

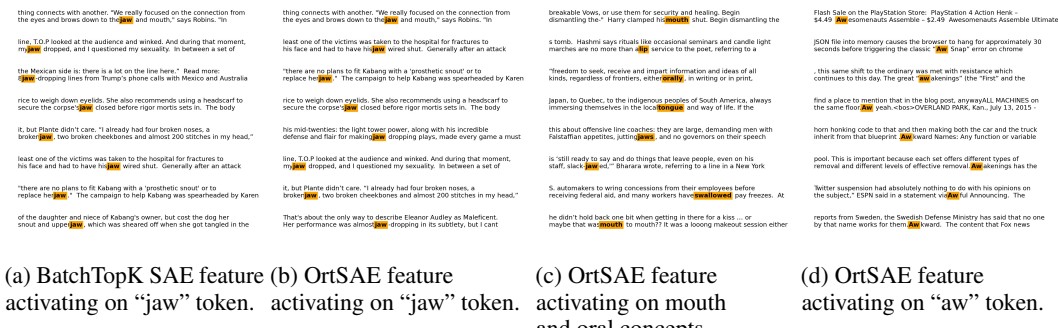

(a) BatchTopK SAE feature activating on "jaw" token.

(b) OrtSAE feature activating on "jaw" token.

(c) OrtSAE feature activating on mouth and oral concepts.

(d) OrtSAE feature activating on "aw" token.

Figure 9: **Decomposition of Jaw-Related BatchTopK SAE Feature into OrtSAE Features.** (a) A BatchTopK SAE (L0=70) feature that activates on the token "jaw", which can be represented as a linear combination of three OrtSAE (L0=70) features: (b) An OrtSAE feature that activates specifically on the token "jaw". (c) An OrtSAE feature that activates on mouth and oral concepts. (d) An OrtSAE feature that activates on the token "aw".

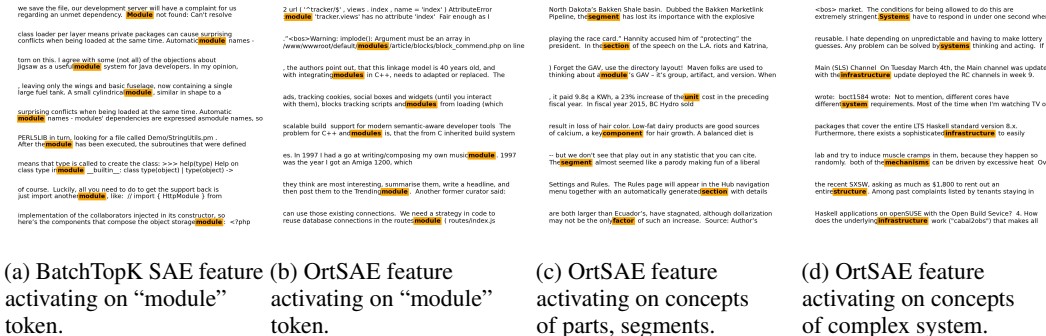

(a) BatchTopK SAE feature activating on "module" token.

(b) OrtSAE feature activating on "module" token.

(c) OrtSAE feature activating on concepts of parts, segments.

(d) OrtSAE feature activating on concepts of complex system.

Figure 10: **Decomposition of Module-Related BatchTopK SAE Feature into OrtSAE Features.** (a) A BatchTopK SAE (L0=70) feature that activates on the token "module", which can be represented as a linear combination of three OrtSAE (L0=70) features: (b) An OrtSAE feature that activates specifically on the token "module". (c) An OrtSAE feature that activates on concepts of parts and segments. (d) An OrtSAE feature that activates on concepts of complex systems.

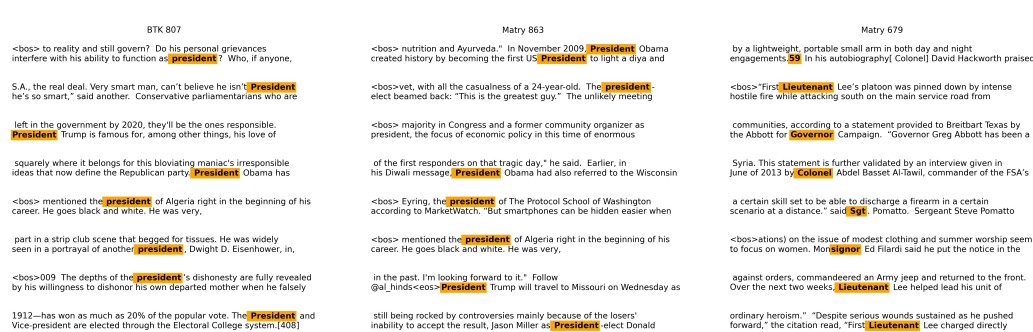

(a) BatchTopK SAE feature activating on president-related terms.

(b) Matryoshka SAE feature activating on president-related terms.

(c) Matryoshka SAE feature activating on military position terms.

Figure 11: **Decomposition of President-Related BatchTopK SAE Feature into Matryoshka SAE Features.** (a) A BatchTopK SAE (L0=70) feature that activates on president-related terms, which can be represented as a linear combination of two Matryoshka SAE (L0=70): (b) A Matryoshka SAE feature that activates specifically on president-related terms. (c) A Matryoshka SAE feature that activates on military position terms.

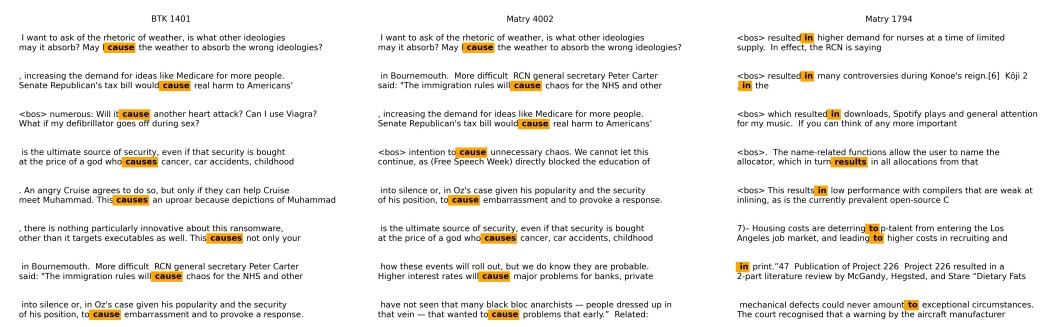

(a) BatchTopK SAE feature activating on cause-related terms.

(b) Matryoshka SAE feature activating on cause-related terms.

(c) Matryoshka SAE feature activating in "effect" contexts

Figure 12: **Decomposition of Cause-Related BatchTopK SAE Feature into Matryoshka SAE Features.** (a) A BatchTopK SAE (L0=70) feature that activates on cause-related terms, which can be represented as a linear combination of two Matryoshka SAE (L0=70): (b) An Matryoshka SAE feature that activates specifically on cause-related terms. (c) An Matryoshka SAE feature that activates in "effect" contexts.

## J  SYNTHETIC EXPERIMENT WITH KNOWN GROUND-TRUTH FEATURES

We designed a synthetic experiment with known ground-truth features to directly evaluate SAEs' ability to recover underlying concepts, following methodologies similar to Chanin and Garriga-Alonso (2025) and Elhage et al. (2022).

We created a synthetic dataset with the following parameters:

1. **Activation dimension**: $n = 100$
2. **Number of ground-truth features**: $m_{\text{true}} = 3200$ (corresponding to an expansion factor of ×32, similar to our Gemma-2-2B experiments)
3. **Dictionary size for SAEs**: $m = 3200$ (matching ground-truth)
4. **Target sparsity for SAEs**: $K = 20$ active features per sample (matching ground-truth)

The data generation process follows:

1. **Feature vectors**: Each ground-truth feature vector $\mathbf{W}_i^{\text{true}}$ is sampled uniformly from the unit sphere $S^{n-1}$.
2. **Feature probabilities**: Each ground-truth feature $\mathbf{W}_i^{\text{true}}$ is assigned an activation probability $p_i = 10^z$ where $z \sim \text{Uniform}(-5, -1.2)$, following a heavy-tailed distribution that mimics real SAEs. This also yields an expected sparsity of 20 active features per sample.
3. **Feature coefficients**: When active, a feature's coefficient $c_i$ is sampled from Log-Normal($\mu = 0, \sigma = 0.25$).
4. **Synthetic activations**: For each sample, we generate independent Bernoulli random variables $b_i \sim \text{Bernoulli}(p_i)$ for $i = 1, \ldots, m_{\text{true}}$. Then the synthetic activation $\mathbf{x}$ is given by:

$$\mathbf{x} = \sum_{i=1}^{m_{\text{true}}} b_i \cdot c_i \cdot \mathbf{W}_i^{\text{true}}. \tag{7}$$

We trained OrtSAE, BatchTopK SAE, Matryoshka SAE, and MP-SAE on this synthetic data using the same training procedures described in Section 4.1. The core evaluation metric was the average cosine similarity between each ground-truth feature vector and its closest learned SAE latent:

$$\text{Recovery Score} = \frac{1}{m_{\text{true}}} \sum_{i=1}^{m_{\text{true}}} \max_{j=1}^{m} \cos(\mathbf{W}_i^{\text{true}}, \mathbf{W}_j^{\text{dec}}) \tag{8}$$

We also evaluated reconstruction quality through Explained Variance and feature orthogonality through MeanCosSim (see 4.2).

Table 5: **Ground-truth feature recovery on synthetic data.** OrtSAE achieves superior recovery of true features while maintaining better reconstruction quality and improved feature orthogonality.

| SAE Method | Recovery Score | Explained Variance | MeanCosSim |
|---|---|---|---|
| BatchTopK SAE | 0.41 | 0.71 | 0.35 |
| Matryoshka SAE | 0.35 | 0.70 | 0.25 |
| MP-SAE | 0.46 | **0.92** | 0.42 |
| OrtSAE | **0.57** | 0.76 | **0.15** |

As shown in Table 5, OrtSAE achieves superior ground-truth feature recovery (0.57 vs. 0.41/0.35/0.46) while maintaining the best feature orthogonality. This demonstrates that orthogonality constraints not only improve feature separation but actively guide SAEs toward the true generative features, providing direct evidence that geometric constraints enhance feature identifiability in superposition. The synthetic experiment validates that OrtSAE's approach fundamentally improves dictionary learning by recovering cleaner, more atomic representations.

## K   GEOMETRIC FOUNDATION OF FEATURE COMPOSITION AND ABSORPTION

In feature composition, a composite feature C ("red square") is learned (by definition) with decoder vector:

$$\mathbf{W}_C^{\text{dec}} = \alpha \mathbf{W}_A^{\text{dec}} + \beta \mathbf{W}_B^{\text{dec}} \tag{9}$$

where $\alpha, \beta > 0$ are positive coefficients.

Both original features A ("red") and B ("square") remain present in the dictionary. Moreover, the cosine similarity between atomic features A and B is bounded by:

$$|\cos(\mathbf{W}_A^{\text{dec}}, \mathbf{W}_B^{\text{dec}})| \leq \epsilon \tag{10}$$

where $\epsilon \approx 0$, since the *superposition hypothesis* (Elhage et al., 2022) posits that neural networks efficiently pack more features than dimensions by representing them in nearly orthogonal directions in activation space.

We now show that feature composition significantly increases cosine similarity:

$$\cos(\mathbf{W}_C^{\text{dec}}, \mathbf{W}_A^{\text{dec}}) \gg \epsilon \tag{11}$$

The cosine similarity between C and A is:

$$\cos(\mathbf{W}_C^{\text{dec}}, \mathbf{W}_A^{\text{dec}}) = \frac{\langle \alpha \mathbf{W}_A^{\text{dec}} + \beta \mathbf{W}_B^{\text{dec}}, \mathbf{W}_A^{\text{dec}} \rangle}{||\mathbf{W}_C^{\text{dec}}|| \, ||\mathbf{W}_A^{\text{dec}}||} = \frac{\alpha ||\mathbf{W}_A^{\text{dec}}||^2 + \beta \langle \mathbf{W}_B^{\text{dec}}, \mathbf{W}_A^{\text{dec}} \rangle}{||\mathbf{W}_C^{\text{dec}}|| \, ||\mathbf{W}_A^{\text{dec}}||} \tag{12}$$

Since $\langle \mathbf{W}_B^{\text{dec}}, \mathbf{W}_A^{\text{dec}} \rangle = \cos(\mathbf{W}_A^{\text{dec}}, \mathbf{W}_B^{\text{dec}})||\mathbf{W}_A^{\text{dec}}|| \, ||\mathbf{W}_B^{\text{dec}}|| \geq -\epsilon ||\mathbf{W}_A^{\text{dec}}|| \, ||\mathbf{W}_B^{\text{dec}}||$ (using the lower bound), we have:

$$\cos(\mathbf{W}_C^{\text{dec}}, \mathbf{W}_A^{\text{dec}}) \geq \frac{\alpha ||\mathbf{W}_A^{\text{dec}}||^2 - \beta \epsilon ||\mathbf{W}_A^{\text{dec}}|| \, ||\mathbf{W}_B^{\text{dec}}||}{||\mathbf{W}_C^{\text{dec}}|| \, ||\mathbf{W}_A^{\text{dec}}||} = \frac{\alpha ||\mathbf{W}_A^{\text{dec}}|| - \beta \epsilon ||\mathbf{W}_B^{\text{dec}}||}{||\mathbf{W}_C^{\text{dec}}||} \tag{13}$$

Now, we compute the norm of $\mathbf{W}_C^{\text{dec}}$ exactly:

$$||\mathbf{W}_C^{\text{dec}}||^2 = ||\alpha \mathbf{W}_A^{\text{dec}} + \beta \mathbf{W}_B^{\text{dec}}||^2 = \alpha^2 ||\mathbf{W}_A^{\text{dec}}||^2 + \beta^2 ||\mathbf{W}_B^{\text{dec}}||^2 + 2\alpha\beta \langle \mathbf{W}_A^{\text{dec}}, \mathbf{W}_B^{\text{dec}} \rangle \tag{14}$$

Using the bound $\langle \mathbf{W}_A^{\text{dec}}, \mathbf{W}_B^{\text{dec}} \rangle \geq -\epsilon ||\mathbf{W}_A^{\text{dec}}|| \, ||\mathbf{W}_B^{\text{dec}}||$, we get:

$$||\mathbf{W}_C^{\text{dec}}||^2 \leq \alpha^2 ||\mathbf{W}_A^{\text{dec}}||^2 + \beta^2 ||\mathbf{W}_B^{\text{dec}}||^2 + 2\alpha\beta\epsilon ||\mathbf{W}_A^{\text{dec}}|| \, ||\mathbf{W}_B^{\text{dec}}|| \tag{15}$$

Taking square roots:

$$||\mathbf{W}_C^{\text{dec}}|| \leq \sqrt{\alpha^2 ||\mathbf{W}_A^{\text{dec}}||^2 + \beta^2 ||\mathbf{W}_B^{\text{dec}}||^2 + 2\alpha\beta\epsilon ||\mathbf{W}_A^{\text{dec}}|| \, ||\mathbf{W}_B^{\text{dec}}||} \tag{16}$$

Therefore:

$$\cos(\mathbf{W}_C^{\text{dec}}, \mathbf{W}_A^{\text{dec}}) \geq \frac{\alpha ||\mathbf{W}_A^{\text{dec}}|| - \beta \epsilon ||\mathbf{W}_B^{\text{dec}}||}{\sqrt{\alpha^2 ||\mathbf{W}_A^{\text{dec}}||^2 + \beta^2 ||\mathbf{W}_B^{\text{dec}}||^2 + 2\alpha\beta\epsilon ||\mathbf{W}_A^{\text{dec}}|| \, ||\mathbf{W}_B^{\text{dec}}||}} \tag{17}$$

For typical feature norms where $||\mathbf{W}_A^{\text{dec}}|| \approx ||\mathbf{W}_B^{\text{dec}}||$, and with $\alpha \approx \beta$ and $\epsilon \approx 0.01$, this simplifies to:

$$\cos(\mathbf{W}_C^{\text{dec}}, \mathbf{W}_A^{\text{dec}}) \geq \frac{\alpha - \alpha \cdot 0.01}{\sqrt{\alpha^2 + \alpha^2 + 2\alpha^2 \cdot 0.01}} = \frac{0.99\alpha}{\sqrt{2.02\alpha^2}} = \frac{0.99}{\sqrt{2.02}} \approx 0.695 \gg 0.01 \tag{18}$$

This proves that feature composition necessarily creates decoder vectors with cosine similarity substantially higher than the near-orthogonal baseline of atomic features. The same logic applies to feature absorption, where a specific feature absorbs a general feature, leading to similar increases in cosine similarity through direct vector inclusion.

## L  THRESHOLD SENSITIVITY ANALYSIS

We evaluated the sensitivity of our cross-model feature overlap analysis across multiple similarity thresholds (0.1-0.4). As shown in Table 6, OrtSAE consistently maintains substantially more unique features than BatchTopK SAE regardless of threshold choice, confirming the robustness of our findings.

Table 6: **Proportion of unique features for different similarity thresholds at L0=70**

| Model | threshold = 0.1 | threshold = 0.2 | threshold = 0.3 | threshold = 0.4 |
|---|---|---|---|---|
| BatchTopK SAE | 0.00% | 1.19% | 3.19% | 7.76% |
| OrtSAE | 2.59% | 8.89% | 11.01% | 15.33% |

## M  BABEL SCORE ANALYSIS

To comprehensively evaluate dictionary coherence beyond pairwise cosine similarity, we employ the Babel score (Tropp, 2004), which quantifies potential interference between feature groups. The Babel function $\mu(k)$ measures the maximum cumulative coherence between any feature and its $k$ most correlated neighbors:

$$\mu(k) = \max_{|\Lambda|=k} \max_{j \notin \Lambda} \sum_{i \in \Lambda} |\cos(\mathbf{W}_i^{\text{dec}}, \mathbf{W}_j^{\text{dec}})| \tag{19}$$

Lower Babel scores indicate better dictionary properties, with ideal dictionaries showing slowly growing $\mu(k)$.

We evaluate coherence at two levels: globally across the entire feature dictionary, and locally among co-activating features. Local coherence is computed per input: for each of 10,000 Gemma-2-2B activation, we calculate interference metrics exclusively among the features that fire together, then average across all activations.

Table 7: **Global and Local Coherence Metrics at L0=70**. Lower values indicate reduced feature interference.

| Method | Local Metrics | | | | Global Metrics | | | |
|---|---|---|---|---|---|---|---|---|
| | MeanCosSim | $\mu(1)$ | $\mu(5)$ | $\mu(20)$ | MeanCosSim | $\mu(1)$ | $\mu(5)$ | $\mu(20)$ |
| BatchTopK SAE | 0.18 | 0.43 | 1.36 | 2.81 | 0.34 | 0.99 | 3.87 | 13.18 |
| Matryoshka SAE | 0.15 | 0.36 | 1.10 | 2.15 | 0.17 | 0.99 | 3.62 | 7.34 |
| OrtSAE | **0.13** | **0.31** | **0.94** | **2.10** | **0.12** | 0.99 | **2.59** | **7.84** |

Table 7 shows OrtSAE achieves superior (lower) scores across all metrics. OrtSAE's local coherence ($\mu(5) = 0.94$) indicates significantly less interference among co-activating features compared to BatchTopK SAE ($\mu(5) = 1.36$) and Matryoshka SAE ($\mu(5) = 1.10$), enabling cleaner feature separation during inference. Globally, OrtSAE creates a well-separated dictionary structure ($\mu(20) = 7.84$ vs 13.18 for BatchTopK SAE).

These improved Babel scores confirm OrtSAE features exhibit less interference during co-activation, enabling more reliable interpretations and superior performance on feature-isolation tasks. These findings align with sparse coding theory (Tropp, 2004; Sigg et al., 2012; Pope et al., 2013), where low mutual coherence guarantees unique sparse recovery. OrtSAE's orthogonality penalty creates

a dictionary approaching these ideal properties, explaining its enhanced atomicity and downstream performance through minimized interference in both global and local contexts.

# N MP-SAE ANALYSIS

Our OrtSAE method and MP-SAE (Costa et al., 2025) differ fundamentally in orthogonality scope: OrtSAE enforces global orthogonality by penalizing pairwise cosine similarities across the entire feature dictionary, while MP-SAE achieves local orthogonality through sequential inference that selects features orthogonal to previously activated ones.

**Why Conditional Orthogonality May Encourage Non-Atomic Features**  This architectural difference has profound implications for feature learning. MP-SAE creates a form of local orthogonality among co-activated features while allowing global correlation structure in the dictionary. As evidenced by Figure 7 in the original MP-SAE paper, their dictionaries exhibit very low global orthogonality (much lower than traditional SAEs) while achieving high local orthogonality. This creates a specific incentive: the model can learn features with high cosine similarity (e.g., a general feature "starts with E" and a composite feature "Elephant + starts with E") as long as they do not frequently co-activate. We hypothesize that this actually creates an ideal environment for the feature absorption and composition problems our paper aims to solve. For instance, the recursive, residual-based inference of MP-SAE might preferentially select the composite feature "Elephant + starts with E" for the token "elephant," completely bypassing the need to ever activate a pure "starts with E" feature. This effectively absorbs the broader concept into more specific, token-aligned features.

**Empirical Evidence**  Our MP-SAE experiment on Gemma-2-2B (L0=25) confirms these concerns: MP-SAE achieves a composition rate of 0.474, significantly higher than all other methods (Table 8). This demonstrates that local orthogonality not only fails to prevent feature composition but actively exacerbates it, as the sequential inference process favors composite features that efficiently minimize residual error at the expense of global atomicity.

Table 8: **Composition rate comparison including MP-SAE at L0=25**

| SAE Method | Explained Variance | Composition rate | MeanCosSim |
|---|---|---|---|
| ReLU SAE | 0.702 | 0.463 | 0.394 |
| BatchTopK SAE | 0.780 | 0.373 | 0.410 |
| Matryoshka SAE | 0.754 | 0.375 | 0.255 |
| OrtSAE | 0.771 | 0.375 | 0.147 |
| MP-SAE | 0.799 | 0.474 | 0.652 |

**The Computational Burden of a Full Comparison**  Beyond the feature quality issues, the sequential encoding process of MP-SAE presents substantial computational challenges for large-scale applications. MP-SAE requires approximately L0/2 forward steps to achieve a target sparsity of L0, making it computationally prohibitive for the sparsity levels we study. For example, training MP-SAE at L0=100 would be roughly 50 times slower than training a traditional SAE.

**Global Orthogonality Naturally Improves the Local Orthogonality**  While applying a conditional orthogonality loss is an interesting direction, we believe it would have a limited effect on the core issues of absorption and composition, as these problems are fundamentally rooted in the global structure of the feature dictionary. Interestingly, although our method uses a global orthogonality penalty, its benefits extend to the local context; As shown in App. M, OrtSAE also achieves the best local Babel score, indicating that promoting global orthogonality naturally leads to improved local orthogonality where it matters, directly mitigating the interference that causes poor atomicity.

