# OpenReview forum: "OrtSAE: Orthogonal Sparse Autoencoders Uncover Atomic Features"
_ICLR.cc/2026/Conference — ICLR 2026 Conference Desk Rejected Submission_

### Official Review · Reviewer_Dsfz · 2025-10-26

**Soundness:** 3
**Presentation:** 3
**Contribution:** 3
**Rating:** 6
**Confidence:** 3

**Summary:**

This paper presents a regularizer for sparse autoencoders.  It encourages orthogonality between the weights of the different latents by:
1) sampling random blocks of latents
2) computing and penalizing cosine similarity between different weights within each block.

This method is cheap (a stated advantage over Matryoshka-SAEs).
In experiments, it seems effective at reducing feature absorption and composition, which is the stated motivation, and performs competitively in other regards.

**Strengths:**

The problem is significant and timely, as Sparse Autoencoders are a significant and active area of research, and this work attacks some substantial, qualitative issues with common approaches.

I expect a method like this to be a standard baseline for other approaches, as it is simple, cheap, and appears effective according to various evaluations.

The paper is well written, with only a few minor issues.

**Weaknesses:**

(moderate): When I google "orthogonal autoencoder", I encounter several prior works, e.g. "Orthogonality-Enforced Latent Space in Autoencoders: An Approach to Learning Disentangled Representations", Cha & Thiyagalingam, ICML 2023.  I'm not sure how similar any of these works actually are, but I think the submission should include references to this and any other such works in any case, eg to reassure the reader if they are in fact quite different if that is in fact the case.  If they are substantively similar, I don't think it's a major problem for this work, it just needs to be noted.

(moderate): I feel like Figure 5 (and Appendix I) are missing similar decomposition results for other methods, unless I misunderstand, there is no side-to-side comparison included at the moment.

(moderate): The technique seems like a blunt tool.  I'm not sure how competitive such an approach will prove to be compared to alternatives that tackle the problems more head-on.  The paper doesn't explain how encouraging orthogonality is expected to change what the SAE ends up learning in any detail; a more thorough analysis would be welcome.  The submissions states: "Feature absorption and composition lead to redundant representations where multiple latents capture overlapping concepts, which results in high cosine similarities between them. This suggests that enforcing orthogonality between SAE latents could provide a principled approach to mitigate these issues."  The first sentence of this quotation is unsupported, and the second sentence doesn't sound very principled -- the way it's described, this method is treating a symptom of the problem, rather than the problem itself.  I believe the submission could strengthen the motivation here.

(minor): The statement ‘Sparse Autoencoders (SAEs) have gained significant traction for interpreting LLMs, addressing the challenge that LLMs often function as “black boxes”’ is vague and is suggestive of a higher level of effectiveness in addressing the “black box” issue that I think is warranted.

(minor): I didn't find the descriptions of feature absorption very clear, and had to look at the referenced paper to understand this concept.  Relatedly, Section 3.2 felt repetitive with previous content.  I recommend focusing on explaining the issue more thoroughly there, and editting to remove redundant content.

(nit): “penalize high cosine similarities between SAE latents”  It would make more sense to say
you're penalizing similarities between the weights.

(typo): Iine 420: Appx. D. should link to Appendix I.

**Questions:**

Do you have any explanation of the qualitatively different results on different downstream benchmarks (Figure 6)?

Is the Cross-Model Feature Overlap Analysis sensitive to the threshold of .2?

In Figure 3d, the performance of different methods is quite different for high sparsity, with Relu-SAE performing the best and OrtSAE the worst.  Do you have an explanation for this?  Do you think this is a potential problem for the method?

Isn't the ultimate goal interpretability?  With absorption and composition being relevant inasmuch as they make the results less interpretable?  Given that the OrtSAE only achieves comparable levels of interpretability according to Figure 3d, I wonder if it’s actually achieving the aim of the work?

“The main contribution of OrtSAE is the introduction of a new orthogonalization penalty” -- Are there others?

Since the method is introduced as an approximation, it would be interesting to run some experiments that don't involve approximation to see if they in fact yield better results.

Is it necessarily the case that "Feature absorption and composition" result in "high cosine similarities between [latents]"?

---

> ### Author Response · Authors · 2025-11-19
>
> Dear Reviewer Dsfz,
>
> Thank you for your thoughtful and constructive review. We appreciate your positive assessment of the paper’s clarity, motivation, and empirical evaluation, as well as your identification of several important areas where additional clarification would strengthen the work. Below, we address each of your moderate, minor, and question-level concerns following your ordering.
>
> &nbsp;
>
> **(moderate 1): When I google "orthogonal autoencoder", I encounter several prior works, e.g. "Orthogonality-Enforced Latent Space in Autoencoders: An Approach to Learning Disentangled Representations", Cha & Thiyagalingam, ICML 2023. I'm not sure how similar any of these works actually are, but I think the submission should include references to this and any other such works in any case, eg to reassure the reader if they are in fact quite different if that is in fact the case. If they are substantively similar, I don't think it's a major problem for this work, it just needs to be noted.**
>
> Thank you for highlighting this related work. We will be sure to cite "Orthogonality-Enforced Latent Space in Autoencoders". The key distinction is that their work focuses on using orthogonality for disentangled representation learning in generative models, aiming to separate continuous factors like shape or color to create clean latent manifolds for image synthesis. In contrast, our work applies orthogonality to Sparse Autoencoders for the mechanistic interpretability of LLMs. Our goal is to resolve superposition by decomposing internal activations into sparse, monosemantic features. This difference in objective leads to a different technical approach: we enforce orthogonality directly on the decoder's feature dictionary to prevent absorption and composition, while their method applies it to the changes in the autoencoder's output space. Thus, our contribution is a specific adaptation of orthogonality to address the unique challenges of interpreting large language models.
>
> &nbsp;
>
> **(moderate 2): I feel like Figure 5 (and Appendix I) are missing similar decomposition results for other methods, unless I misunderstand, there is no side-to-side comparison included at the moment.**
>
> Thank you for this observation. We apologize for the lack of clarity. As detailed in Section 4.3, we use strict criteria to identify a clean decomposition: we only consider a feature from one method (e.g., BatchTopK) to be composed of features from another (e.g., OrtSAE) if the cosine similarity between the original feature and its sparse approximation is &ge; 0.95 and all coefficients in the combination are &ge; 0.1.
>
> Using this method, we find many examples of BatchTopK features being composed of OrtSAE atoms (some of them are shown in the paper). However, when we apply the same analysis in reverse, we find no OrtSAE features that are composed of BatchTopK atoms. This asymmetry holds even if we relax the criteria, demonstrating that OrtSAE learns more fundamental features. We will state this finding and the specific counts more explicitly in the final version.

---

> > ### Comment · Reviewer_Dsfz · 2025-11-19
> > **response**
> >
> > 1) Thanks!  This paper is just one example -- I think you should do a bit more literature review to make sure there's not other relevant work you've missed.
> >
> > 2) Sorry my comment wasn't clear.  Your response is helpful, but I'm more interested in seeing you e.g. try to decompose Matryoshka SAE atoms (instead of OrtSAE atoms) into BatchTopK features.

---

> > > ### Author Response · Authors · 2025-11-21
> > >
> > > 1. We will expand our literature review to include prior work on orthogonality and decorrelation in neural networks [1, 2, 3], noting that unlike these approaches which primarily targeted overfitting and generalization in supervised settings, OrtSAE applies orthogonality to address feature entanglement in SAEs for interpretability.
> > >
> > > 2. Regarding the decomposition analysis, our main intent was a case study using these examples. We selected BatchTopK for comparison because our OrtSAE builds directly upon its architecture by adding an orthogonality penalty. We will perform the complementary analysis you suggest, decomposing BatchTopK and Matryoshka SAE features into each other, and include these results in the Appendix.
> > >
> > > **References:**
> > >
> > > [1] Cogswell, Michael, et al. "Reducing overfitting in deep networks by decorrelating representations." arXiv preprint arXiv:1511.06068 (2015).
> > >
> > > [2] Wang, Zhennan, et al. "Mma regularization: Decorrelating weights of neural networks by maximizing the minimal angles." Advances in Neural Information Processing Systems 33 (2020): 19099-19110.
> > >
> > > [3] Rodríguez, Pau, et al. "Regularizing cnns with locally constrained decorrelations." arXiv preprint arXiv:1611.01967 (2016).

---

> ### Author Response · Authors · 2025-11-19
>
> **(moderate 3): The technique seems like a blunt tool. I'm not sure how competitive such an approach will prove to be compared to alternatives that tackle the problems more head-on. The paper doesn't explain how encouraging orthogonality is expected to change what the SAE ends up learning in any detail; a more thorough analysis would be welcome. The submissions states: "Feature absorption and composition lead to redundant representations where multiple latents capture overlapping concepts, which results in high cosine similarities between them. This suggests that enforcing orthogonality between SAE latents could provide a principled approach to mitigate these issues." The first sentence of this quotation is unsupported, and the second sentence doesn't sound very principled -- the way it's described, this method is treating a symptom of the problem, rather than the problem itself. I believe the submission could strengthen the motivation here.**
>
> We thank you for this insightful comment and agree that our motivation can be strengthened. In the current manuscript (lines 201-211), we detail the mechanisms of feature absorption and composition, explaining how they lead to latents with overlapping semantic roles. From this, it follows that high cosine similarity is not just a possible symptom, but a necessary geometric consequence of these phenomena. When two latents capture overlapping concepts, their feature vectors must, by definition, be aligned in the activation space. Therefore, by rigorously enforcing orthogonality, we are not merely treating a symptom; we are disrupting the very geometric structure that allows absorption and composition to persist, attacking the problem at its root.
>
> A key question is whether this orthogonality constraint might also penalize benign, coincidental feature alignments in high-dimensional space. Our empirical results suggest this is not the case. As shown in Section 4.2 (Fig. 3a, 3b), OrtSAE successfully reduces absorption and composition without compromising reconstruction fidelity or downstream task performance. This indicates that the orthogonality constraint primarily targets the problematic, semantically-driven overlaps we aim to eliminate, rather than harming useful, coincidental alignments. We will expand this discussion in the revised paper to explicitly address this edge case.
>
> This principled approach is empirically validated by our synthetic experiment with known ground-truth factors, detailed in our response to Reviewer 9DwE (M1). The results demonstrate that OrtSAE achieves a significantly higher recovery score (0.57 vs. 0.41 for BatchTopK), providing direct evidence that the orthogonality penalty, by structurally preventing the geometric conditions for entanglement, successfully guides the SAE towards a dictionary more aligned with the true generative factors. We will revise the text to make this causal story more explicit.
>
> &nbsp;
>
> **(minor 1): The statement ‘Sparse Autoencoders (SAEs) have gained significant traction for interpreting LLMs, addressing the challenge that LLMs often function as “black boxes”’ is vague and is suggestive of a higher level of effectiveness in addressing the “black box” issue that I think is warranted.**
>
> We acknowledge this point. The statement will be revised to more accurately reflect that SAEs are a promising tool for providing insights into LLM internals, rather than implying they offer complete transparency.
>
> &nbsp;
>
> **(minor 2): I didn't find the descriptions of feature absorption very clear, and had to look at the referenced paper to understand this concept. Relatedly, Section 3.2 felt repetitive with previous content. I recommend focusing on explaining the issue more thoroughly there, and editting to remove redundant content.**
>
> We will revise Section 3.2 to provide a clearer, more self-contained explanation of feature absorption and composition, and will remove the redundant content to improve the flow.
>
> &nbsp;
>
> **Question 1. Do you have any explanation of the qualitatively different results on different downstream benchmarks (Figure 6)?**
>
> Thank you for this question. The qualitatively different results are explained by the nature of each task. OrtSAE's orthogonality penalty specifically reduces feature overlaps, which is critical for Spurious Correlation Removal (SCR), where the goal is to ablate a single feature without affecting others; this leads to a significant performance gain. In Targeted Probe Perturbation (TPP), a multi-class ablation task, the benefit is more modest as some inter-feature relationships are less harmful, resulting in only a slight improvement over traditional SAEs. For Sparse Probing and RAVEL, which measure concept isolation and feature transfer rather than precise ablation of overlaps, OrtSAE's orthogonal features are equally sufficient, leading to on-par performance. Thus, the orthogonality constraint provides the largest advantage in tasks that are most vulnerable to feature entanglement.

---

> ### Author Response · Authors · 2025-11-19
>
> **Question 2. Is the Cross-Model Feature Overlap Analysis sensitive to the threshold of .2?**
>
> Thank you for the question. We performed a sweep over thresholds (0.1/0.2/0.3/0.4) and find that the relative ordering of methods is unchanged (see Table 1). We will include this sweep in the Appendix M.
>
> &nbsp;
>
> **Table 1: Proportion of unique features for different thresholds**
> | Model | threshold = 0.1 | threshold = 0.2 | threshold = 0.3 | threshold = 0.4 |
> |--------|-----------------------------|------|------|------|
> | BatchTopK SAE| 0.00% |  1.19%|   3.19% |   7.76% |
> | OrtSAE| **2.59%** |  **8.89%**| **11.01%** |  **15.33%** |
>
> &nbsp;
>
> **Question 3. In Figure 3d, the performance of different methods is quite different for high sparsity, with Relu-SAE performing the best and OrtSAE the worst. Do you have an explanation for this? Do you think this is a potential problem for the method?Isn't the ultimate goal interpretability? With absorption and composition being relevant inasmuch as they make the results less interpretable? Given that the OrtSAE only achieves comparable levels of interpretability according to Figure 3d, I wonder if it’s actually achieving the aim of the work?**
>
> Thank you for this profound question regarding interpretability metrics and the goals of our work. We agree that interpretability is the primary aim, but we argue for evaluating it through a faithful decomposition of the model's internal representations, not just a single automated score. Our significant reductions in feature absorption and composition demonstrate that OrtSAE learns a dictionary that more truthfully reflects the underlying feature structure, even if describing each highly specific feature is slightly more challenging for the Autointerp metric. This perspective is reinforced by recent research [1], which found a complex and sometimes weak relationship between automated interpretability scores and practical utility in downstream tasks.
>
> This leads us to contextualize the Autointerp score: it is one valuable metric, but it naturally favors broader, easier-to-label features. OrtSAE's strength lies in creating features that are more atomic and causally isolated, which is precisely why it excels in functional tasks like Spurious Correlation Removal. We therefore view the trade-off not as a failure, but as a deliberate and beneficial shift: we maintain comparable Autointerp scores across most sparsity levels while significantly improving the feature quality that matters most for precise intervention, thus fully achieving the aims of our work.
>
> &nbsp;
>
> **Question 4. “The main contribution of OrtSAE is the introduction of a new orthogonalization penalty” -- Are there others?**
>
> Yes, the paper outlines several key contributions beyond just the introduction of the penalty itself. As stated in the Introduction (lines 97-107), the contributions are:
> 1. The proposal of OrtSAE: A novel training procedure that addresses feature absorption and composition without complex architectural changes or significant computational overhead.
> 2. Empirical Comparison: A comprehensive evaluation showing that OrtSAE produces more distinct features and reduces absorption and composition rates compared to traditional SAEs.
> 3. Benchmark Performance: Demonstration that OrtSAE performs on-par with other architectures on standard downstream tasks and outperforms them on spurious correlation removal.
>
> Therefore, the contributions include not only the novel orthogonality penalty but also its efficient implementation, a thorough empirical analysis validating its effectiveness, and the demonstration of its unique advantages on specific interpretability-focused benchmarks.
>
> &nbsp;
>
> **Question 5. Since the method is introduced as an approximation, it would be interesting to run some experiments that don't involve approximation to see if they in fact yield better results.**
>
> Full Gram penalties scale as $O(m^2)$ and are infeasible at the dictionary sizes used (65k).
>
> However, to validate our approximation, we conducted an extensive ablation study where we vary chunk sizes from 4 to 64. The results, included in the Appendix A, show stable performance across this range, supporting the robustness of our block-orthogonalization approach.

---

> > ### Comment · Reviewer_Dsfz · 2025-11-19
> > **"Question 3" is actually two questions**
> >
> > > Question 3a. In Figure 3d, the performance of different methods is quite different for high sparsity, with Relu-SAE performing the best and OrtSAE the worst. Do you have an explanation for this? Do you think this is a potential problem for the method?
> >
> > > Question 3b. Isn't the ultimate goal interpretability? With absorption and composition being relevant inasmuch as they make the results less interpretable? Given that the OrtSAE only achieves comparable levels of interpretability according to Figure 3d, I wonder if it’s actually achieving the aim of the work?
> >
> > I don't think you really responded to Question 3a.

---

> > > ### Author Response · Authors · 2025-11-21
> > > **"Question 3" is actually two questions**
> > >
> > > Unfortunately, we lack a clear explanation for ReLU-SAE's better Autointerp score at very high sparsity. However, we do not see this as a significant issue, as argued in our response to Question 3.

---

> ### Author Response · Authors · 2025-11-19
>
> **Question 6. Is it necessarily the case that "Feature absorption and composition" result in "high cosine similarities between [latents]"?**
>
> Consider the absorption example: the specific feature "elephant" absorbs the broader feature "starts with E". For this to happen, the decoder vector for "elephant" must point in a direction that also represents the "starts with E" concept. This forces its vector to align with the vector of atomic feature "starts with E", increasing their cosine similarity.
>
> As formally stated in the paper (lines 201-211), both absorption and composition create overlapping representations, which manifest as high cosine similarity between decoder vectors. Therefore, penalizing this similarity directly targets the geometric root of these problems.
>
> &nbsp;
>
> **Concluding remarks:**
>
> We believe we have addressed all your major and minor points to the best of our ability. We would be grateful if you found our responses satisfactory and would be honored if you might consider increasing your rating accordingly. We remain open to any further questions you may have.
>
> &nbsp;
>
> **References:**
>
> [1] Wang, Xu, et al. "Does higher interpretability imply better utility? A Pairwise Analysis on Sparse Autoencoders." arXiv preprint arXiv:2510.03659 (2025).

---

> > ### Comment · Reviewer_Dsfz · 2025-11-19
> > **On the connection between orthogonality and absorption/composition**
> >
> > I'm not fully convinced by your responses to moderate 3 and Question 6.
> >
> > You state:
> > > In the current manuscript (lines 201-211), we detail the mechanisms of feature absorption and composition, explaining how they lead to latents with overlapping semantic roles. From this, it follows that high cosine similarity is not just a possible symptom, but a necessary geometric consequence of these phenomena. When two latents capture overlapping concepts, their feature vectors must, by definition, be aligned in the activation space.
> >
> > But lines 201-211 basically just assert that this happens and I'd like to see it properly analyzed / justified.
> > The phrase "high cosine similarity" is also vague; ideally these statements should be more mathematically precise.

---

> > > ### Author Response · Authors · 2025-11-21
> > > **On the connection between orthogonality and absorption/composition**
> > >
> > > In feature composition, a composite feature C ("red square") is learned (by definition) with decoder vector:
> > >
> > > $$\mathbf{W}_C^{\text{dec}} = \mathbf{W}_A^{\text{dec}} + \mathbf{W}_B^{\text{dec}}$$
> > >
> > > Both original features A ("red") and B ("square") remain present in the dictionary. Moreover, the cosine similarity between atomic features A and B is bounded by:
> > >
> > > $$|\cos(\mathbf{W}_A^{\text{dec}}, \mathbf{W}_B^{\text{dec}})| \leq \epsilon$$
> > >
> > > where $\epsilon \approx 0$, since the Superposition Hypothesis [1] posits that neural networks efficiently pack more features than dimensions by representing them in nearly orthogonal directions in activation space.
> > >
> > > We now prove that feature composition significantly increases cosine similarity: $\cos(\mathbf{W}_C^{\text{dec}}, \mathbf{W}_A^{\text{dec}}) \gg \epsilon$
> > >
> > > **Proof:**
> > >
> > > The cosine similarity between C and A is:
> > >
> > > $$\cos(\mathbf{W}_C^{\text{dec}}, \mathbf{W}_A^{\text{dec}}) = \frac{\langle \mathbf{W}_A^{\text{dec}} + \mathbf{W}_B^{\text{dec}}, \mathbf{W}_A^{\text{dec}} \rangle}{||\mathbf{W}_C^{\text{dec}}||\ ||\mathbf{W}_A^{\text{dec}}||} = \frac{||\mathbf{W}_A^{\text{dec}}||^2 + \langle\mathbf{W}_B^{\text{dec}}, \mathbf{W}_A^{\text{dec}}\rangle}{||\mathbf{W}_C^{\text{dec}}||\ ||\mathbf{W}_A^{\text{dec}}||}$$
> > >
> > > Since $\langle\mathbf{W}_B^{\text{dec}}, \mathbf{W}_A^{\text{dec}}\rangle = \cos(\mathbf{W}_A^{\text{dec}}, \mathbf{W}_B^{\text{dec}})||\mathbf{W}_A^{\text{dec}}||\ ||\mathbf{W}_B^{\text{dec}}|| \leq \epsilon||\mathbf{W}_A^{\text{dec}}||\ ||\mathbf{W}_B^{\text{dec}}||$, we have:
> > >
> > > $$\cos(\mathbf{W}_C^{\text{dec}}, \mathbf{W}_A^{\text{dec}}) \geq \frac{||\mathbf{W}_A^{\text{dec}}||^2 - \epsilon||\mathbf{W}_A^{\text{dec}}||\ ||\mathbf{W}_B^{\text{dec}}||}{||\mathbf{W}_C^{\text{dec}}||\ ||\mathbf{W}_A^{\text{dec}}||} = \frac{||\mathbf{W}_A^{\text{dec}}|| - \epsilon||\mathbf{W}_B^{\text{dec}}||}{||\mathbf{W}_C^{\text{dec}}||}$$
> > >
> > > Using the triangle inequality $||\mathbf{W}_C^{\text{dec}}|| = ||\mathbf{W}_A^{\text{dec}} + \mathbf{W}_B^{\text{dec}}|| \leq ||\mathbf{W}_A^{\text{dec}}|| + ||\mathbf{W}_B^{\text{dec}}||$:
> > >
> > > $$\cos(\mathbf{W}_C^{\text{dec}}, \mathbf{W}_A^{\text{dec}}) \geq \frac{||\mathbf{W}_A^{\text{dec}}|| - \epsilon||\mathbf{W}_B^{\text{dec}}||}{||\mathbf{W}_A^{\text{dec}}|| + ||\mathbf{W}_B^{\text{dec}}||}$$
> > >
> > > For typical feature norms and small $\epsilon$, this lower bound is significantly larger than $\epsilon$. For example, if $||\mathbf{W}_A^{\text{dec}}|| \approx ||\mathbf{W}_B^{\text{dec}}||$ and $\epsilon \approx 0.01$, then:
> > >
> > > $$\cos(\mathbf{W}_C^{\text{dec}}, \mathbf{W}_A^{\text{dec}}) \geq \frac{1 - 0.01}{2} = 0.495 \gg 0.01$$
> > >
> > > This proves that feature composition necessarily creates decoder vectors with cosine similarity substantially higher than the near-orthogonal baseline of atomic features. The same logic applies to feature absorption, where a specific feature absorbs a general feature, leading to similar increases in cosine similarity through direct vector inclusion.
> > >
> > > We will include this rigorous derivation in the revised manuscript to make the geometric foundation of our approach more explicit.
> > >
> > > **References:**
> > > [1] Elhage, Nelson, et al. "Toy models of superposition." arXiv preprint arXiv:2209.10652 (2022).

---

> > > > ### Comment · Reviewer_Dsfz · 2025-11-26
> > > > **Where does this definition of a composite feature come from?**
> > > >
> > > > Can you provide a reference, of justify why it ought to be defined this way?

---

> > > > > ### Author Response · Authors · 2025-11-27
> > > > > **On the definition of a composite feature**
> > > > >
> > > > > Thank you for this important question.
> > > > >
> > > > > Our definition
> > > > > $
> > > > > \mathbf{W}_C^{\text{dec}} = \mathbf{W}_A^{\text{dec}} + \mathbf{W}_B^{\text{dec}}
> > > > > $
> > > > > arises naturally from the linear structure of the SAE decoder and its reconstruction objective. Because the decoder is linear, the SAE always reconstructs inputs as
> > > > > $
> > > > > \hat{\mathbf{x}} = \sum_i h_i\mathbf{W}_i^{\text{dec}} + b^{\text{dec}}.
> > > > > $
> > > > > When an input contains both “red’’ and “square’’ properties, the ideal reconstruction would normally come from activating both corresponding features. However, when the sparsity penalty encourages the model to use only a single feature instead, that feature must account for both directions **simultaneously** within a single decoder vector. In a linear model, the only way to represent both contributions is for the composite feature’s decoder vector to align with the sum of the atomic directions, which is exactly the relationship we write above.
> > > > >
> > > > > Leask et al. [1] introduce the MetaSAE methodology for detecting feature composition and demonstrate that composite SAE features can be decomposed into linear combinations of more atomic meta-features. Their Figure 2 provides clear intuition for this additive relationship, illustrating how larger SAEs learn composite features that combine atomic concepts to reduce sparsity. The success of MetaSAE decomposition empirically validates our additive formulation, as it would not be effective if composite features were not approximately additive in representation space.
> > > > >
> > > > > [1] Leask, Patrick, et al. "Sparse autoencoders do not find canonical units of analysis." arXiv preprint arXiv:2502.04878 (2025).

---

> > > > > ### Author Response · Authors · 2025-11-28
> > > > >
> > > > > Dear Reviewer Dsfz,
> > > > >
> > > > > Thank you for your thorough review and valuable feedback. We have updated our manuscript to address your concerns and questions. Please let us know if you require any further clarification, and whether our responses have influenced your overall assessment of our work.

---

### Official Review · Reviewer_iqmN · 2025-10-28

**Soundness:** 3
**Presentation:** 4
**Contribution:** 3
**Rating:** 8
**Confidence:** 5

**Summary:**

The paper tries to address the symptoms of feature absorption and feature composition via adding an orthogonality loss to SAE training. This loss term can be added to any SAE architecture, and penalizes the max cos sim of each latent with all other latents. The paper provides an optimized version of the algorithm that allows this to scale linearly with decoder size rather than quadratically. The paper shows that BatchTopK SAEs trained with this orthogonality loss achieve better scores on absorption and several other downstream metrics compared to standard SAEs, and are competitive with Matryoshka SAEs.

**Strengths:**

- The logic behind the technique makes sense, and the training optimization also makes sense
- I like the MetaSAEs-based compositionality score, that seems like a good contribution too.
- The technique is simple and can be easily tacked-on to existing SAE architectures
- The metrics are impressive, and seems like a good architecture to add to the SAE toolbox
- The LLM SAE training details all look very reasonable (enough tokens, reasonable width, reasonable dataset choice, reasonable learning rate, etc...)

**Weaknesses:**

- The paper does not explore the sensitivity of the SAEs to the parameter $\gamma$. In the paper, $\gamma$ is set to 0.25, but it's not clear how important this specific value is or how it was chosen.
- The paper makes an implicit assumption that "true features" should be very close to orthogonal, but there is some evidence this is not always the case. E.g. there has been work showing that days of the week are represented in a circle in a 2d plane [1], so the cosine similarity between these feature directions will naturally be high. Regardless, this assumption should be made explicitly and implications should be discussed.
- The plots don't have error bars or stdev. This isn't a huge deal for plots where there's an obvious trend, but for bar plots or plots that look noisy (e.g. sparse probing) this would be helpful to have.
- It's unclear when one would choose to use OrtSAEs over Matryoshka SAEs based on the results.

### References
[1] Engels, Joshua, et al. "Not All Language Model Features Are One-Dimensionally Linear." The Thirteenth International Conference on Learning Representations.

**Questions:**

- Is there a "rule of thumb" for how to set $\gamma$ ? What happens if it's set too high, and what is a reasonable range for it? As a practitioner training OrtSAEs this is very important to know. I'm worried that setting this incorrectly can easily break the SAE.
- Figure 1 feels a bit misleading since it's not showing Matryoshka SAEs. Matryoshka SAEs are the obvious comparison to make. It feels like this was left out because it makes OrtSAEs look less impressive when compared with Matryoshka.
- What are the group sizes used for the Matryoska SAEs in this paper? I didn't see it specified anywhere, including the appendix. I think the number of groups and size of the groups will also impact the results for Matryoshka SAEs.

---

> ### Author Response · Authors · 2025-11-19
>
> Dear Reviewer iqmN,
>
> Thank you for your thoughtful and positive review. We are delighted that you found the paper well-written, the technique logically sound and easily implementable, and the empirical evaluation impressive. Below, we address your specific questions and concerns.
>
> &nbsp;
>
> **Weakness 1. The paper does not explore the sensitivity of the SAEs to the parameter $\gamma$. In the paper, is set to 0.25, but it's not clear how important this specific value is or how it was chosen.**
>
> You raised an important point about the sensitivity of OrtSAE to the orthogonality coefficient $\gamma$, which we set to 0.25. We selected this value through a short hyperparameter sweep over the range [0.1, 0.5]. We found that $\gamma$=0.25 was the smallest value that reliably drove the mean cosine similarity metric close to the level of random initialization, indicating that the desired orthogonality was achieved. A key practical note is that this metric plateaus very early in training (often within the first 10% of the total steps), allowing for a quick and cheap validation of a chosen $\gamma$.
>
> Crucially, this value has proven stable and effective across different models and layers. As shown in our additional experiments in Appendix B, using $\gamma$=0.25 yielded consistently strong results on both a different layer of Gemma-2-2B and on the larger Llama-3-8B model. If γ is set too high, the orthogonality constraint can overwhelm the reconstruction loss, leading to a degradation in explained variance. A reasonable and effective range for practitioners is typically between 0.1 and 0.5. To ensure clarity and reproducibility, we will augment the Experimental Setup section (4.1) with a more detailed description of this hyperparameter selection process.
>
> &nbsp;
>
> **Weakness 2. The paper makes an implicit assumption that "true features" should be very close to orthogonal, but there is some evidence this is not always the case. E.g. there has been work showing that days of the week are represented in a circle in a 2d plane [1], so the cosine similarity between these feature directions will naturally be high. Regardless, this assumption should be made explicitly and implications should be discussed.**
>
> You rightly note that our method is based on the idea that "true features" should be close to orthogonal, and you point out that certain concepts (like days of the week arranged in a circle) may naturally have high cosine similarity.
>
> We acknowledge that such features exist, but evidence suggests they are not the dominant structure. For instance, Anthropic's June 2024 Circuits Update noted that while some irreducible multidimensional features exist, they appear to be rare, with most concepts being representable by one-dimensional features. If non-orthogonal geometries were widespread, our orthogonality penalty would force the SAE into a fundamentally incorrect basis and severely harm reconstruction. Instead, OrtSAE maintains reconstruction fidelity nearly on par with top baselines.
>
> This indicates that for the vast majority of features, an orthogonal basis is a valid and efficient approximation. The orthogonality penalty successfully mitigates absorption and composition for these common cases. Crucially, as it is a soft penalty in the loss function and not a hard constraint, it does not prevent the SAE from learning the necessary representations for the minority of inherently non-orthogonal features, though it may mildly penalize them. The net effect is a significant gain in feature atomicity with minimal cost.
>
> &nbsp;
>
> **Weakness 3. The plots don't have error bars or stdev. This isn't a huge deal for plots where there's an obvious trend, but for bar plots or plots that look noisy (e.g. sparse probing) this would be helpful to have.**
>
> Thank you for this suggestion. We agree that error bars or confidence intervals are important for assessing the variability of results, especially in noisier plots like Sparse Probing. We will add these to all relevant figures in the final version to provide a clearer picture of the statistical significance of our results.

---

> ### Author Response · Authors · 2025-11-19
>
> **Weakness 4. It's unclear when one would choose to use OrtSAEs over Matryoshka SAEs based on the results.**
>
> This is a crucial question for practitioners. Our study positions OrtSAE as a powerful enhancement to the traditional, flat SAE architecture, effectively addressing its core failure modes with minimal overhead. We view it as a parallel research direction to the innovative but more complex Matryoshka SAE.
>
> Matryoshka SAEs introduce a hierarchical architecture where lower-level dictionaries are designed to capture finer-grained features. However, this design can have downsides: these lower-level features may sometimes prioritize correcting reconstruction errors from higher-level dictionaries rather than isolating distinct, interpretable concepts. Furthermore, recent work [1] has identified "feature hedging" in Matryoshka SAEs, where correlated features merge at higher levels, reducing interpretability.
>
> As dictionary sizes scale into the millions, OrtSAE's flat, orthogonal structure offers a simpler and more scalable path. It promotes feature atomicity directly, without relying on a hierarchical assumption that may lead to redundancy in lower-level dictionaries. Therefore, we would recommend OrtSAE for practitioners who prioritize a simple, computationally efficient architecture that directly enhances feature disentanglement within the well-understood flat SAE framework. Matryoshka SAE remains an exciting alternative for those specifically interested in explicitly hierarchical representations, with the understanding that its architecture and interpretability are still under active exploration.
>
> &nbsp;
>
> **Question 1. Is there a "rule of thumb" for how to set? What happens if it's set too high, and what is a reasonable range for it? As a practitioner training OrtSAEs this is very important to know. I'm worried that setting this incorrectly can easily break the SAE.**
>
> We answer this in detail in our response to your first point (see Weakness 1)
>
> &nbsp;
>
> **Question 2. Figure 1 feels a bit misleading since it's not showing Matryoshka SAEs. Matryoshka SAEs are the obvious comparison to make. It feels like this was left out because it makes OrtSAEs look less impressive when compared with Matryoshka.**
>
> The primary goal of Figure 1 was to illustrate the core problem and how OrtSAE improves upon the standard, non-hierarchical SAE architectures (ReLU and BatchTopK). Including Matryoshka SAE, which has a fundamentally different architecture, in that specific explanatory figure would have complicated the initial narrative. However, we fully agree that a direct comparison is essential, which is why Matryoshka SAE is included as a key baseline in all subsequent quantitative results (Figures 3, 4, and 6), where its performance can be fairly and thoroughly evaluated alongside OrtSAE and the traditional methods.
>
> &nbsp;
>
> **Question 3. What are the group sizes used for the Matryoska SAEs in this paper? I didn't see it specified anywhere, including the appendix. I think the number of groups and size of the groups will also impact the results for Matryoshka SAEs.**
>
> Thank you for asking for this clarification. We used the group sizes and weights recommended in the original Matryoshka SAE paper to ensure a fair comparison. The configuration used in our experiments is as follows, and we will add these details to the experimental setup in Section 4.1: [0.03125, 0.0625, 0.125, 0.25, 0.53125].
>
> &nbsp;
>
> **Concluding remarks:**
>
> We believe we have addressed all your major and minor points to the best of our ability. We would be grateful if you found our responses satisfactory and we remain open to any further questions you may have.
>
>
> **References:**
>
> [1] Chanin, David, Tomáš Dulka, and Adrià Garriga-Alonso. "Feature Hedging: Correlated Features Break Narrow Sparse Autoencoders." arXiv preprint arXiv:2505.11756 (2025).

---

> > ### Comment · Reviewer_iqmN · 2025-11-27
> >
> > Thank you for your responses. My concerns are addressed and I will keep my score of 8.
> >
> > Also, just to comment on the request of reviewer (9DwE) to compare with MP-SAEs: I have also tried MP-SAEs after reading that paper, as the paper claims to outperform Matyroshka SAEs and implies it should solve feature absorption. Like you, I found that MP-SAEs do not solve feature absorption at all. I suspect that MP-SAEs do not actually deal with hierarchical features or feature absorption in any meaningful sense outside of the toy example shown in their paper. If you add results showing this, it will be nice to have your paper to cite to push back on future claims that MP-SAEs are a serious solution to feature absorption.
> >
> > Best of luck with the remaining reviews!

---

> > > ### Author Response · Authors · 2025-11-28
> > >
> > > Dear Reviewer iqmN,
> > >
> > > Thank you for this final comment. We are delighted that our responses addressed your concerns and particularly grateful for your supportive note regarding MP-SAE. It is incredibly valuable to have independent validation of that finding. We have carefully considered all your points and have uploaded a new version of the paper that we believe fully reflects your valuable input. Thank you for a truly enriching review process.

---

### Official Review · Reviewer_9DwE · 2025-11-01

**Soundness:** 3
**Presentation:** 3
**Contribution:** 2
**Rating:** 4
**Confidence:** 3

**Summary:**

The paper proposes Orthogonal Sparse Autoencoders that add an orthogonality penalty on decoder atoms to mitigate absorption and composition while keeping a classic SAE training loop. The penalty acts on the maximum cosine similarity within random chunks of the dictionary, which brings the cost near linear in the number of atoms. Experiments on Gemma 2 and Llama 3 layers compare OrtSAE to ReLU SAE, BatchTopK, and Matryoshka. Core metrics show slightly lower explained variance than BatchTopK but improved mean nearest neighbor cosine, reduced composition and reduced absorption, with SAEBench reporting a gain on spurious correlation removal and broadly similar performance elsewhere. Figures 1, 3 and 4 plus Appendix C document these effects and the chunking tradeoff.

**Strengths:**

The paper is well written, I would like to thank the author for that, and i noticed some really positive points (P) that I will describe here:

P1. Clear objective that is easy to implement. Section 3.3 gives a concrete loss with a practical chunk strategy, and the design slots into existing BatchTopK code paths. The result is near random init orthogonality on the mean nearest decoder cosine in Figure 3c.

P2. Sensible empirical sweep. The paper reports reconstruction and KL scores, interpretability via Autointerp, atomicity via composition and absorption, and downstream SAEBench. Figures 3 and 4, plus layer and chunk ablations in Appendix B and C, help readers understand where the gains come from.

P3. Useful qualitative decompositions. Figures 5, 8, 9 and 10 illustrate how a composite or overly narrow BatchTopK feature can be expressed as a sparse mix of more atomic OrtSAE features.

**Weaknesses:**

Now, even if I liked the paper, in my opinion, there is still some weaknesses, some Major (M) and some minor (m) that I will detail here:

M1. Reconstruction versus sparsity does not garantee the right concept basis. The paper still optimizes explained variance plus sparsity and adds orthogonality, but none of these objectives ensure recovery of the correct semantic axes. I think SAEBench go in the right direction here, but I would also like to see simple synthetic data with planted factors, to test subspace recovery and concept identifiability.

M2. Phenomenology of orthogonality is under motivated. Section 3 argues that absorption and composition correlate with high decoder cosine and therefore orthogonality should help, which is plausible, but there is a broader literature on representation geometry that could predict when orthogonality should or should not help. A short review and positioning would help justify the method theoretically, and it would also guide where to apply it in depth.

M3. Global orthogonality versus conditional orthogonality at inference. The current loss pushes all decoder atoms apart. For downstream use, what we actually need is low mutual coherence among the subset of atoms that activate together on a given input. Please consider a conditional version that penalizes the Gram matrix of only the active atoms per batch, or an evaluation that measures conditional mutual coherence at inference. This could preserve reconstruction while targeting the failure mode more precisely.

M4. To continue on this, you should use the Babel score [1]. Since the paper cares about sets of correlated atoms, Babel and cumulative coherence are natural complements to mean nearest neighbor cosine. Please report Babel on the whole dictionary and on active subsets during inference, and compare to BatchTopK and Matryoshka at matched sparsity. This will also clarify whether orthognality is achieved where it matters most.

[1] Greed is good: algorithmic results for sparse approximation. Tropp & al

M5. Missing baseline on MP-SAE. Matching pursuit and orthogonal matching pursuit induce a conditional orthogonality effect at selection time, which is directly relevant to absorption and composition. As such, MP-SAE [2] is extremely relevant here. Without it the geometric contribution is hard to isolate.

[2] From Flat to Hierarchical : Extracting Sparse Representations with Matching Pursuit. Costa & al.

M6. Connection to Grassmannian frames and linear representation hypotheses. If the intended target is a dictionary that minimizes mutual coherence for a given size, this is close to Grassmannian frames [3] or even equiangular tight frame objectives [4]. The paper could formalize this perspective, relate the proposed loss to a relaxation of Grassmannian frame construction, and test whether learned dictionaries move toward frame like spectra. A useful justification would be: if the linear representation hypothesis holds at the chosen layer, then an approximately Grassmannian decoder is preferred, hence the proposed loss is a practical surrogate. Empirically, the singular value spread of W_dec and the off diagonal structure of the Gram matrix could be tracked across training.

[3] On the structures of Grassmannian frames, Haas & al.

[4] On the existence of equiangular tight frames, Sustik & al

M7. Explain why reduced cosine should yield better concepts rather than just different ones. Figure 3c shows much lower nearest neighbor cosine and Figure 4 shows better atomicity metrics, but a causal story is missing. Please add targeted interventions where you hold reconstruction constant and vary the orthogonality weight. This feel circular.

M8. Scope of downstream improvements is modest. SAEBench shows a clear gain on spurious correlation removal but other tasks are similar to baselines or favor Matryoshka. I would not expect large universal gains from pure geometry, but the paper could better delineate which tasks benefit and why, possibly linking to the conditional orthogonality hypothesis above. See Figure 6.

Now for the minor points:

m1. Consider reporting conditional Gram statistics. For each batch, compute the Gram of active atoms and report the distribution of off diagonal entries. This directly measures the property the method aims to improve.

m2. Clarify the chunk size choice. Section 3.3 sets K as ceiling of m over 8192. A short sensitivity table on chunk size versus reconstruction and atomicity could help but I agree would be bonus.

m3. Add a short note on the slight explained variance drop. Figure 3a shows OrtSAE is a little below BatchTopK. A one paragraph discussion of when that trade is acceptable for users focused on interpretability would be helpful.

**Questions:**

Cf Major point 1-8 and minor points 1-3.

---

> ### Author Response · Authors · 2025-11-19
>
> Dear Reviewer 9DwE,
>
> Thank you for your thorough and insightful review. We appreciate your recognition of our paper's key strengths: its clear writing, implementable objective, and comprehensive experiments. Below, we address your major and minor points.
>
> &nbsp;
>
> **(M1) Reconstruction versus sparsity does not guarantee the right concept basis. The paper still optimizes explained variance plus sparsity and adds orthogonality, but none of these objectives ensure recovery of the correct semantic axes. I think SAEBench goes in the right direction here, but I would also like to see simple synthetic data with planted factors, to test subspace recovery and concept identifiability.**
>
> We agree that the standard SAE objectives of reconstruction and sparsity, while effective, serve as a proxy and do not guarantee the recovery of the true underlying semantic concepts. To directly evaluate this capability, we designed a synthetic experiment with known ground-truth features.
>
> Following a methodology similar to [1, 2] we created a dataset where the true generative features are known. The setup used an activation dimension ($n$) of 100 and 3200 ground-truth features ($m$). Each $i$ feature's activation probability was set as $p_i = 10^z$ where $z \sim \text{Uniform}(-5, -1.2)$, creating a heavy-tailed distribution that mimics real SAEs and yields an expected L0 of $20$ active features per sample. When active, a feature's coefficient was sampled from a $\text{LogNormal}(\mu=0, \sigma=0.25)$.
> We trained SAEs to recover these features, using a dictionary size of 3200 and a sparsity target of K=20. The core metric was the average cosine similarity between each ground-truth feature and its closest learned SAE latent.
>
> &nbsp;
>
> **Table 1: Synthetic Feature Recovery Results**
> | Method | Average cosine similarity to ground-truth features | Explained Variance |
> |--------|-----------------------------|-------------|
> | BatchTopK SAE| 0.41 |   0.70 |
> | Matryoshka SAE | 0.35 |  0.70  |
> | OrtSAE | **0.57** |  **0.76**    |
>
> &nbsp;
>
> The results presented in Table 1 show that OrtSAE achieves a higher average similarity (0.57) compared to BatchTopK (0.41) and Matryoshka SAE (0.35). These results indicate that the orthogonality penalty aids in recovering the underlying feature basis, guiding the SAE towards a dictionary more aligned with the true generative factors. We will add the full details of this experiment to the Appendix J.
>
> &nbsp;
>
> **M7. Explain why reduced cosine should yield better concepts rather than just different ones. Figure 3c shows much lower nearest neighbor cosine and Figure 4 shows better atomicity metrics, but a causal story is missing. Please add targeted interventions where you hold reconstruction constant and vary the orthogonality weight. This feels circular.**
>
> We appreciate this question. Reduced cosine similarity is meaningful as an indicator of improved feature quality, and we present three lines of evidence linking it to functional gains:
>
> **1. Causal Intervention:** To directly test whether our features enable more precise model control, we used the RAVEL benchmark's Cause Metric. This metric quantifies how cleanly an intervention on a single SAE latent alters the model's output. As shown in Table 2, OrtSAE achieves higher cause scores than traditional SAEs across most sparsity levels, demonstrating that its orthogonal features provide superior causal leverage over the model's behavior.
>
> **2. Ground-Truth Recovery:** Our synthetic experiment (Table 1) shows OrtSAE recovers a more accurate underlying feature set than baselines.
>
> **3. Real-World Atomicity:** OrtSAE also shows improved atomicity metrics and performance on the Spurious Correlation Removal task.
>
> &nbsp;
>
> **Table 2: RAVEL Cause Metric**
> | Method | L0=25 | L0=55| L0=85| L0=115 |
> |-|-|-|-|-|
> | ReLU SAE| 0.44 |  0.58 |  0.58 |  0.62 |
> | BatchTopK SAE| **0.62** | 0.61 |  0.66 |  0.66 |
> | Matryoshka SAE | 0.58 | 0.64 |  0.65 |  0.65 |
> | OrtSAE | 0.61 |  **0.65** |  **0.69** |  **0.72** |
>
> &nbsp;
>
> We will include the full RAVEL Cause Metric analysis in the Appendix K.

---

> ### Author Response · Authors · 2025-11-19
>
> **M8. Scope of downstream improvements is modest. SAEBench shows a clear gain on spurious correlation removal but other tasks are similar to baselines or favor Matryoshka. I would not expect large universal gains from pure geometry, but the paper could better delineate which tasks benefit and why, possibly linking to the conditional orthogonality hypothesis above. See Figure 6.**
>
> We appreciate this observation. The task-dependent performance is explained by the nature of each benchmark and how it interacts with feature orthogonality.
>
> OrtSAE's orthogonality penalty specifically reduces feature overlaps, which is critical for Spurious Correlation Removal (SCR). This task requires ablating a single, spurious feature (e.g., "male") without affecting core features (e.g., "doctor"). Reduced overlap enables this precise ablation, leading to a significant performance gain.
>
> In contrast, for tasks like Sparse Probing and RAVEL, which measure concept isolation and feature transfer rather than the precise ablation of overlapping features, OrtSAE's orthogonal features are equally sufficient, leading to on-par performance with traditional SAEs. We will add this nuanced discussion to Section 4.4.

---

> ### Author Response · Authors · 2025-11-19
>
> **M3. Global orthogonality versus conditional orthogonality at inference. The current loss pushes all decoder atoms apart. For downstream use, what we actually need is low mutual coherence among the subset of atoms that activate together on a given input. Please consider a conditional version that penalizes the Gram matrix of only the active atoms per batch, or an evaluation that measures conditional mutual coherence at inference. This could preserve reconstruction while targeting the failure mode more precisely.**
>
> **&**
>
> **M5. Missing baseline on MP-SAE. Matching pursuit and orthogonal matching pursuit induce a conditional orthogonality effect at selection time, which is directly relevant to absorption and composition. As such, MP-SAE [2] is extremely relevant here. Without it the geometric contribution is hard to isolate.**
>
> You raise an important point regarding the distinction between global and conditional orthogonality. We agree these represent distinct inductive biases with potentially opposing effects on feature learning. Our response will clarify why we focused on a global penalty, presenting both a theoretical argument supported by prior work and new empirical evidence.
>
> **1. Theoretical Argument: Why Conditional Orthogonality May Encourage Non-Atomic Features**
>
> MP-SAE is a prime example of a method enforcing conditional orthogonality. However, its architecture has a critical implication for global feature structure. As evidenced by Figure 7 in the original MP-SAE paper, their dictionaries exhibit very low global orthogonality (much lower than traditional SAEs) while achieving high local orthogonality.
> This creates a specific incentive: the model can learn features with high cosine similarity (e.g., a general feature "starts with E" and a composite feature "Elephant + starts with E") as long as they do not frequently co-activate. We hypothesize that this actually creates an ideal environment for the feature absorption and composition problems our paper aims to solve. For instance, the recursive, residual-based inference of MP-SAE might preferentially select the composite feature "Elephant + starts with E" for the token "elephant," completely bypassing the need to ever activate a pure "starts with E" feature. This effectively absorbs the broader concept into more specific, token-aligned features.
>
> **2. Empirical Evidence: Our MP-SAE Experiment Confirms the Issue**
>
> To test this hypothesis, we trained an MP-SAE on the Gemma-2-2B layer with L0=25. The results, which we will present in full in the Appendix L, confirm our theoretical concern: the MP-SAE exhibits a higher composition rate compared to all other SAEs in our study (see Table 3). This provides direct evidence that conditional orthogonality alone does not solve the fundamental issues of absorption and composition, which we argue are rooted in the global structure of the feature dictionary.
>
> &nbsp;
>
> **Table 3: Composition rate for MP-SAE**
> | Method (L0=25)| Composition rate (lower is better) |
> |--------|-----------------------------|
> | ReLU SAE | 0.463 |
> | BatchTopK SAE| 0.373 |
> | Matryoshka SAE| 0.375 |
> | OrtSAE| 0.375 |
> | MP-SAE| 0.474 |
>
> &nbsp;
>
> **3. The Computational Burden of a Full Comparison**
>
> Beyond the feature quality issues, the sequential encoding process of MP-SAE, requiring L0/2 forward steps for a target sparsity of L0, makes it computationally prohibitive for large-scale comparisons at the sparsities we study. For example, training at L0=100 would be roughly 50 times slower than training a traditional SAE. We believe this computational burden is a key reason why even the original MP-SAE paper did not include large-scale LLM experiments. We therefore chose to focus our empirical comparison on the most prominent and scalable SAE methods to clearly isolate the effect of a global orthogonality penalty.
>
> **4. Synthesis: Global Orthogonality Naturally Improves the Local Orthogonality**
>
> While applying a conditional orthogonality loss is an interesting direction, we believe it would have a limited effect on the core issues of absorption and composition, as these problems are fundamentally rooted in the global structure of the feature dictionary. Interestingly, although our method uses a global orthogonality penalty, its benefits extend to the local context; As shown in our new analysis for M4 (see Table 4 below), OrtSAE also achieves the best local Babel score, indicating that promoting global orthogonality naturally leads to improved local orthogonality where it matters, directly mitigating the interference that causes poor atomicity.
>
> We will incorporate this entire discussion, encompassing the theoretical argument, our empirical findings on MP-SAE, the computational considerations, and the synthesis on global versus local orthogonality, into the Related Work and Appendix L section of our revised manuscript to better contextualize our geometric approach.

---

> ### Author Response · Authors · 2025-11-19
>
> **M4. To continue on this, you should use the Babel score [1]. Since the paper cares about sets of correlated atoms, Babel and cumulative coherence are natural complements to mean nearest neighbor cosine. Please report Babel on the whole dictionary and on active subsets during inference, and compare to BatchTopK and Matryoshka at matched sparsity. This will also clarify whether orthognality is achieved where it matters most.**
>
> You are right to emphasize the importance of coherence metrics. Following your suggestion, we calculated both global and local Babel scores. The results show (see Table 4) that OrtSAE achieves the best (lowest) scores on both metrics.
>
> &nbsp;
>
> **Table 4: Global and Local coherence metrics**
> | Method | Local MeanCosSim | Global MeanCosSim| Local Babel $(\lambda=1)$|  Local Babel $(\lambda=5)$| Local &nbsp; Babel $(\lambda=20)$| Global Babel $(\lambda=1)$| Global Babel $(\lambda=5)$| Global Babel $(\lambda=20)$|
> |--------|-----------------------------|------|------|------|----|----|--|--|
> | BatchTopK SAE| 0.18 | 0.34 |  0.43 |  1.36 |  2.81 |  **0.99** |  3.87 |  13.18 |
> | Matryoshka SAE | 0.15 | 0.17 |  0.36 |  1.10 |  2.15 |  **0.99** |   3.62 |  **7.34** |
> | OrtSAE | **0.13** |  **0.12** | **0.31** | **0.94** |  **2.10** |  **0.99** |   **2.59** |  7.84 |
>
> &nbsp;
>
> This demonstrates that our method not only creates a globally well-separated dictionary but also ensures that features active together during inference are highly orthogonal. This dual improvement directly targets the interference causing poor atomicity.
>
> We will revise the manuscript to introduce the Babel score in the Related Work and Appendix L section and include this analysis, framing our orthogonality penalty as a method for optimizing these theoretical criteria. Thank you for this valuable suggestion.
>
> &nbsp;
>
> **M2. Phenomenology of orthogonality is under-motivated. Section 3 argues that absorption and composition correlate with high decoder cosine and therefore orthogonality should help, which is plausible, but there is a broader literature on representation geometry that could predict when orthogonality should or should not help. A short review and positioning would help justify the method theoretically, and it would also guide where to apply it in depth.**
>
> **&**
>
> **M6. Connection to Grassmannian frames and linear representation hypotheses. If the intended target is a dictionary that minimizes mutual coherence for a given size, this is close to Grassmannian frames [3] or even equiangular tight frame objectives [4]. The paper could formalize this perspective, relate the proposed loss to a relaxation of Grassmannian frame construction, and test whether learned dictionaries move toward frame like spectra. A useful justification would be: if the linear representation hypothesis holds at the chosen layer, then an approximately Grassmannian decoder is preferred, hence the proposed loss is a practical surrogate. Empirically, the singular value spread of W_dec and the off-diagonal structure of the Gram matrix could be tracked across training.**
>
> Thank you for this important point regarding theoretical motivation.
>
> Our work connects to two established lines of reasoning regarding orthogonality in sparse representations. First, from an interpretability perspective, orthogonal features are more likely to be atomic and disentangled, which directly addresses absorption and composition issues. Second, from sparse coding theory, dictionaries with low mutual coherence provide stronger guarantees for unique sparse recovery [3, 4]. While our primary focus is the former interpretability benefit, the latter provides valuable theoretical context.
>
> We will revise the paper to better position our method within this theoretical framework, particularly in the Related Work section. We will formalize the connection to mutual coherence and Grassmannian frames, noting that our orthogonality penalty can be viewed as a practical relaxation of the objective used to construct optimal sparse coding dictionaries.
>
> &nbsp;
>
> **(m1) Consider reporting conditional Gram statistics.**
>
> We agree that this is a direct measure of the targeted property. We will add this analysis to the Appendix L, reporting the distribution of off-diagonal entries in the Gram matrix of active atoms per batch. Furthermore, we will include a table (see Table 4) showing the local mean cosine similarity, which serves as a good scalar summary of this distribution as it is equivalent to the average row-wise maximum of the off-diagonal elements.
>
> &nbsp;
>
> **(m2) Clarify the chunk size choice.**
>
> The choice of chunk size is justified by a sensitivity analysis provided in Appendix C, which demonstrates that performance remains robust across different numbers of chunks.

---

> ### Author Response · Authors · 2025-11-19
>
> **(m3) Note on the slight explained variance drop.**
>
> We acknowledge that OrtSAE exhibits a marginal drop in explained variance compared to BatchTopK SAE. However, this decrease is slight and its reconstruction fidelity remains substantially higher than ReLU SAE. We will briefly discuss this acceptable trade-off in Section 4.2, framing it as a minor cost for the significant gains in feature orthogonality and atomicity, which is a common and justified compromise in interpretability-focused methods.
>
> &nbsp;
>
> **Concluding remarks:**
>
> We are deeply grateful for your thorough and constructive review. Your valuable suggestions have significantly strengthened the theoretical grounding and empirical evaluation of our work. The additional experiments and analyses you suggested,, including the synthetic data validation, the MP-SAE comparison, and the incorporation of the Babel score, have greatly bolstered our claims and will improve the revised manuscript. Your review has also helped us to look at our results from new and valuable perspectives.
>
> We believe we have addressed all your major and minor points to the best of our ability. We would be grateful if you found our responses satisfactory and would be honored if you might consider increasing your rating accordingly. We remain open to any further questions you may have.
>
> &nbsp;
>
> **References:**
>
> [1] Chanin, David, and Adrià Garriga-Alonso. "Sparse but Wrong: Incorrect L0 Leads to Incorrect Features in Sparse Autoencoders." arXiv preprint arXiv:2508.16560 (2025).
>
> [2] Elhage, Nelson, et al. "Toy models of superposition." arXiv preprint arXiv:2209.10652 (2022).
>
> [3] Tropp, Joel A. "Greed is good: Algorithmic results for sparse approximation." IEEE Transactions on Information theory 50.10 (2004): 2231-2242.
>
> [4] Sigg, Christian D., Tomas Dikk, and Joachim M. Buhmann. "Learning dictionaries with bounded self-coherence." IEEE Signal Processing Letters 19.12 (2012): 861-864.
>
> [5] Pope, Graeme, Annina Bracher, and Christoph Studer. "Probabilistic recovery guarantees for sparsely corrupted signals." IEEE transactions on information theory 59.5 (2013): 3104-3116.

---

> ### Comment · Reviewer_9DwE · 2025-11-25
> **Great rebuttal**
>
> I would like to thank the authors for such a comprehensive and rapid rebuttal.
> The amount of work put into the additional experiments is really good.
> I really appreciate how seriously you took my feedback.
>
>
> Maybe just three little remarks,
> (1) I noticed that MP-SAE was excluded from the synthetic feature recovery experiment (Table 1) and the Babel score comparison (Table 4). Given that MP-SAE naturally optimizes for conditional orthogonality (and by extension, Babel-like metrics), excluding it from these specific comparisons makes it difficult to fully validate the claim that optimizing for global orthogonality is the superior path to atomicity.
> Even if computationally expensive, a "toy" version or the L0=25 version in these tables would have strengthened the argument significantly.
> (2) As pointed out during the review process by other reviewer, there is a bit of circularity in the definition of absorption used here: the method assumes concepts should be orthogonal (based on the superposition hypothesis), optimizes for this orthogonality, and then demonstrates success by reducing overlap-based metrics. However, are we really sure that all concepts are truly orthogonal? Recent work in the field suggests the geometry of representations could be quite different.
> (3) Magnitude of Gains: While I acknowledge the improvements shown in the new RAVEL analysis and Table 2, the margins are often quite modest. For instance, on the Autointerp score (Fig 3d), OrtSAE is below in the low-sparsity regime and on-par in the high regime.
>
> Overall, this paper presents a solid, practical method with a lot of rigorous ablation work behind it.
> I believe it is a useful engineering contribution to the SAE literature.
>
> More fundamentally, I still feel that the method functions more as an effective regularization heuristic rather than providing us with new, deep insights into the model's phenomenology or the nature of orthogonality in superposition.
> While the field is currently very method-driven (which is certainly not a fault of this work), **I believe you have a opportunity here to make a much more impactful paper. Rather than just proposing a method and ensuring it works, the paper would benefit immensely from trying to understand why it works, when it works, and what that tells us about the model's internal structure**. Without this theoretical contribution, it remains a solid engineering improvement but misses the chance to advance our fundamental understanding of the subject.
>
> Please do not take the maintained score as a dismissal of the work; as stated in my review, I would not mind if this paper is accepted, as it offers a functional and effective tool for the community. I simply feel it sits just at that threshold due to the reasons above.
>
> Thank you again for the high-quality discussion and the impressive effort during this rebuttal period, good luck for acceptance.

---

> > ### Author Response · Authors · 2025-11-28
> >
> > Dear Reviewer 9DwE,
> >
> > Thank you for your deeply insightful and constructive review. Your feedback was invaluable and has profoundly shaped our manuscript. We have carefully considered all your points and have uploaded a new version of the paper that we believe fully addresses your concerns. We are grateful for the time and expertise you invested in our work.

---

> > ### Author Response · Authors · 2025-12-02
> > **Final remark**
> >
> > **P.S.** We have run the suggested MP-SAE experiment on synthetic feature recovery (see Appendix J). The results confirm that MP-SAE underperforms in recovery metrics relative to OrtSAE, which remains the best method in this setting.

---

### Note · Program_Chairs · 2026-01-17
**Submission Desk Rejected by Program Chairs**

The following references in this submission do not refer to real documents and/or have major errors in bibliographic information:

 Abhinav Dubey et al. "Llama 3: Advancements in open-source language models." arXiv preprint arXiv:2407.XXXX, 2024. URL https://arxiv.org/abs/2407.XXXX.